# The Genetic Architecture of the Human Corpus Callosum and its Subregions

Ravi R. Bhatt [1,4] ✉, Shruti P. Gadewar [1,4] ✉, Ankush Shetty[1], Iyad Ba Gari [1], Elizabeth Haddad [1], Shayan Javid [1], Abhinaav Ramesh[1], Elnaz Nourollahimoghadam[1], Alyssa H. Zhu[1], Christiaan de Leeuw [2], Paul M. Thompson[1], Sarah E. Medland [3] & Neda Jahanshad [1] ✉

The corpus callosum (CC) is the largest set of white matter fibers connecting the two hemispheres of the brain. In humans, it is essential for coordinating sensorimotor responses and performing associative or executive functions. Identifying which genetic variants underpin CC morphometry can provide molecular insights into the CC's role in mediating cognitive processes. We developed and used an artificial intelligence based tool to extract the mid-sagittal CC's total and regional area and thickness in two large public datasets. We performed a genome-wide association study (GWAS) meta-analysis of European participants (combined $N = 46,685$) with generalization to the non-European participants (combined $N = 7040$). Post-GWAS analyses implicated prenatal intracellular organization and cell growth patterns, and high heritability in regions of open chromatin. Results suggest programmed cell death mediated by the immune system drives the thinning of the posterior body and isthmus. Genetic overlap, and causal genetic liability, between the CC, cerebral cortex features, and neuropsychiatric disorders such as attention-deficit/hyperactivity, bipolar disorders, and Parkinson's disease were identified.

The corpus callosum (CC) is the largest white matter tract in the human brain, facilitating higher order functions of the cerebral cortex by allowing the two hemispheres of the brain to communicate[1,2]. This connection is essential for coordinating sensorimotor responses, performing associative and executive functions, and representing information in multiple dimensions[3,4]. Most CC fibers connect corresponding left and right cortical regions of the brain, with the organization, development of axonal elongation, and myelination of callosal fibers being correlated with the rostro-caudal (front-to-back) distribution of functional areas[5,6]. Regional alterations in CC shape are easily assessed with neuroimaging studies, which have found local callosal abnormalities in complex neurodevelopmental and neuropsychiatric disorders[6–11], such as, on average, lower anterior volumes in people with autism spectral disorder[12] and lower posterior thickness

in individuals with bipolar disorder[13]. Twin studies show up to 66% heritability for CC area[14,15], and previous single-cohort studies of genetic influences on CC volume and its relationship to neuropsychiatric disorders have found heritability estimates between 22–39%[16,17]. Yet, the interplay between genetic variants influencing CC morphometry, the cerebral cortex, and associated neuropsychiatric disorders is not well understood.

Three-dimensional (3D) magnetic resonance imaging (MRI) provides a non-invasive approach to quantify individual variations in brain regions and their connections[6], including the morphometry of the CC, and how they are associated with brain-based traits and diseases. The midsagittal section of an anatomical brain MRI scan is able to capture the entire rostro-caudal formation of the CC, which is almost always in the field of view of 2D clinical and 3D research MRI scans alike. This 2D

[1]Imaging Genetics Center, Mark and Mary Stevens Neuroimaging and Informatics Institute, Keck School of Medicine, University of Southern California, Marina del Rey, CA, USA. [2]Department of Complex Trait Genetics, Centre for Neurogenomics and Cognitive Research, VU University, Amsterdam, The Netherlands. [3]Psychiatric Genetics, QIMR Berghofer Medical Research Institute, Brisbane 4006, Australia. [4]These authors contributed equally: Ravi R. Bhatt, Shruti P. Gadewar. ✉e-mail: rbhatt@usc.edu; gadewar@usc.edu; njahansh@usc.edu

midsagittal representation can be segmented to offer a lower dimensional projection of the anatomical intricacies of the CC, allowing for structural measures of CC area and thickness to be computed[18–20]. We developed and validated a fully automated artificial intelligence based CC feature extraction tool, *Segment, Measure, and AutoQC the midsagittal CC* (*SMACC*), which we make publicly available at smacc[20].

Using data from the UK Biobank[21] (UKB) and Adolescent Brain Cognitive Development[22] (ABCD) studies, here we present results from a genome-wide association study (GWAS) meta-analysis of total area and mean thickness of the CC derived using *SMACC*. We also present the results for five differentiated areas based on distinguishable projections to (1) prefrontal, premotor, and supplementary motor, (2) motor, (3) somatosensory, (4) posterior parietal and superior temporal, and (5) inferior temporal and occipital cortical brain regions[23,24]. These regions are believed to represent structural-functional coherence[6]. We performed a GWAS meta-analysis using two population-based cohorts, one of adolescents and another of older adults, to examine distinct genetic influences on CC area and thickness[25,26]. The principal analyses were in individuals of European ancestry and the same analyses were then repeated using the data from non-European participants to assess consistency in the magnitude and direction of effect sizes. Downstream post-GWAS analyses investigated the enrichment of genetic association signals in tissue types, cell types, brain regions, and biological pathways. We examined the genetic overlap at the global and local level, using LD Score regression (LDSC)[27] and Local Analysis of Variant Association (LAVA)[28], respectively, and the causal genetic relationships between CC phenotypes, cortical morphometry, and related neuropsychiatric conditions.

## Results

### Characterization of corpus callosum shape associated loci

We conducted a GWAS of area and mean thickness of the whole CC, and five regions of the Witelson parcellation scheme (Fig. 1)[23,24], using data from participants of European ancestry from the UKB ($N = 41,979$) and ABCD cohorts ($N = 4706$). A meta-analysis of GWAS summary statistics of all CC derived metrics in UKB and ABCD was performed using METAL and the random-metal extension[29,30], based on the DerSimonian-Laird random-effects model (**Methods**). To examine the generalizability of single-nucleotide polymorphism (SNP) effects across ancestries, these same analyses were run using data from non-European participants (total $N = 7040$).

The GWAS meta-analysis identified 48 independent significant SNPs for total area and 18 independent SNPs for total mean thickness. Independent significant SNPs were determined in FUMA using the default threshold of $r^2 = 0.6$, and genomic loci were determined at $r^2 = 0.1$. This identified 28 genomic loci for total cross-sectional area, and 11 genomic loci for total mean thickness. All significant loci for total area and mean thickness showed concordance in the direction of effect between the two cohorts. There were 5 loci, all in intronic regions, each positionally mapped to genes[31] that overlapped between area and mean thickness. These included *IQCJ-SHIP1* (multimolecular complexes of initial axon segments and nodes of Ranvier, and calcium mediated responses)[32], *FIP1L1* (RNA binding and protein kinase activity)[33], *HBEGF* (growth factor activity and epidermal growth factor receptor binding)[34], *CDKN2B-AS1* (involved in the NF-κB signaling pathway with diverse roles in the nervous system)[35,36], and *FAM107B* (cytoskeletal reorganization in neural cells and cell migration/expansion)[37]. The genomic locus mapped to *IQCJ-SHIP1* had a positive effect for total area (rs11717303, effect allele: C, effect allele frequency (EAF): 0.689, $\beta = 4.28$, s.e. = 0.51, $p = 4.54 \times 10^{-17}$). The same locus showed a negative effect for a different SNP on total thickness (rs12632564, effect allele: T, EAF: 0.305, $\beta = -0.042$, s.e. = 0.006, $p = 2.59 \times 10^{-12}$). The strongest locus for total area (rs7561572, effect allele: A, EAF: 0.532, $\beta = -4.13$, s.e. = 0.46, $p = 1.98 \times 10^{-18}$) was positionally mapped to the *STRN* gene. The strongest locus for mean thickness (rs4150211, effect allele: A, EAF: 0.265, $\beta = -0.05$, s.e. = 0.006, $p = 8.20 \times 10^{-18}$) was mapped to the *HBEGF* gene.

Loci for area overlapped between parcellations in a rostral-caudal gradient (1–5), such that: rs1122688 on the *SHTN1* (or *KIAA1598*) gene (involved in positive regulation of neuron migration) overlapped

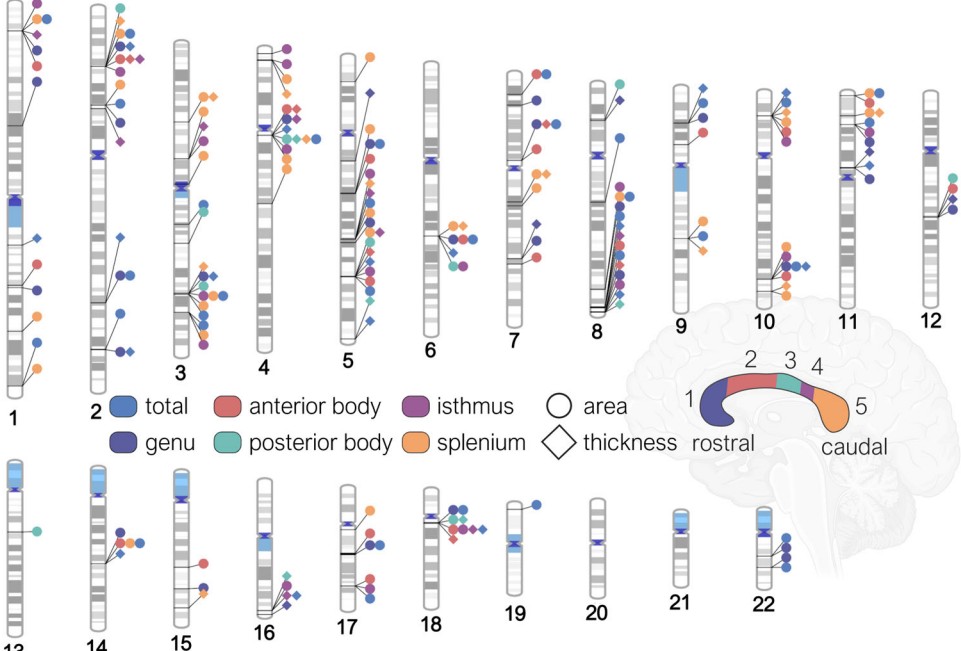

**Fig. 1 | Regions of the midsagittal corpus callosum and associated genomic loci.** An ideogram representing loci that influence total CC area, its mean thickness, and area and thickness of individual parcellations determined by the Witelson parcellation scheme in a rostral-caudal gradient (1–5). Results shown are from an inverse-weighted random-effects meta-analysis (DerSimonian-Laird method). Reported p-values are two-sided. All loci are significant at the Bonferroni corrected, experiment-wide threshold of $p < 6.13 \times 10^{-9}$. Created in part by using Biorender.com (agreement number UX28RS3P2L).

between the genu (1) and anterior body (2); rs1268163 near the *FOXO3* gene (involved in IL-9 signaling and *FOXO*-mediated transcription) overlapped between the posterior body (3) and isthmus (4); and rs11717303 on the *IQCJ-SCHIP1* gene overlapped between the isthmus (4) and splenium (5). This gradient pattern was not observed for mean thickness. The strongest regional association was observed with splenium area (rs10901814, effect allele: C, EAF: 0.584, $\beta = -1.69$, s.e. = 0.16 $p = 2.02 \times 10^{-24}$) and thickness (rs11245344, effect allele: T, EAF: 0.570, $\beta = -0.11$, s.e. = 0.11, $p = 6.28 \times 10^{-22}$), both on the *FAM53B* gene. *FAM53B* is involved in the positive regulation of the canonical Wnt signaling pathway. We observed overlaps in the direction and magnitude of effects between the main European analyses with the results from the non-European participants. 124 (82%) out of the 152 significant loci identified across CC phenotypes in this study had effect sizes in European participants falling within the 95% confidence interval of those seen in the non-European participants. Furthermore, 78 loci demonstrated a consistent direction of effect across all four cohorts (2 European from UKB and ABCD, and 2 non-European from UK Biobank and ABCD, respectively). Detailed annotations and regional association plots of all genomic loci, independent significant SNPs and genes are in Supplementary Data 1–4 and Supplementary Data 42.

In order to test for the influence of total intracranial volume (ICV) on the GWAS, another set of GWASes were run controlling for ICV (Methods, Supplementary Data 37). The overlap coefficients and genetic correlations, respectively, were 0.64 and 0.75 (s.e.= 0.03) for total area and 1 and 1.02 (s.e. = 0.008) for total thickness, indicating a high degree of overlap for both analyses. Near perfect genetic correlations were observed comparing across all CC traits, except splenium area ($r_g = 0.08$, s.e. = 0.05), which may be driven by collider bias[38,39] (Supplementary Data 38). Overall, the strongest enrichment signals were observed among genes that were significant in both the ICV-adjusted and non-ICV GWAS, as well as genes uniquely significant in the non-ICV GWAS - compared to the GWAS that included ICV as a covariate alone. Overall, greater enrichment was observed with genes common between ICV and specific to no ICV, compared to the GWAS which included ICV as a covariate. Genes mapped to significant loci common to both GWAS sets, were consistently mapped to canonical signaling pathways, including PI3K/AKT, PDGFR, and estrogen signaling, suggesting ICV-insensitive mechanisms involved in CC development and maintenance. Genes unique to the no ICV GWAS showed enrichment for mitochondrial respiration, oxidative stress response, and integrin signaling. Genes specific to the ICV-controlled GWAS exhibited much lower enrichment. Regionally specific enrichments were observed for area traits, including WDR5-mediated epigenetic regulation in the genu and apoptosis, as implicated by previous analyses, in the isthmus (Supplementary Data 39).

## SNP heritability and genetic correlation between cohorts

Moderate to high genetic correlations were seen across CC phenotypes between cohorts using LDSC, with $r_g$ ranging from 0.54 (s.e. = 0.27) and 0.92 (s.e. = 0.63) for area metrics, and 0.30 (s.e. = 0.16) and 0.99 (s.e. = 0.69) for thickness metrics. To complement the LDSC approach with an approach using individual level data, we used the bivariate GREML in GCTA[40]. Moderate genetic correlations between cohorts were seen using bivariate GCTA with $r_g$ ranging between 0.40 (s.e. = 0.04) and 0.49 (s.e. = 0.03) across all traits. Age-related variability in white matter likely contributes to some of the lower correlation agreements between cohorts, as white matter volume tends to increase through childhood and adolescence, peak in early adulthood, and then gradually decline from middle age onward[41]. For instance, certain genetic variants might exert a stronger influence on CC structure during periods of white matter growth (as in the younger ABCD cohort) compared to periods of white matter decline (as in the older UKB cohort). The smaller sample size of the ABCD cohort may limit LDSC's ability to detect polygenic effects, capturing primarily the

strongest genetic signals[42]. However, strong cross-cohort correlations for total area and isthmus thickness phenotypes suggest that genetic variants affecting these traits are likely consistent across developmental stages[43,44] to estimate SNP heritability ($h^2_{SNP}$) and generic correlations between each cohort. Within the UKB, heritability values ranged for different CC phenotypes from 0.42 to 0.71, with similar results seen in the ABCD cohort (Supplementary Data 5–8). Total area (UKB $h^2_{SNP}$ = 0.72, s.e. = 0.01; ABCD $h^2_{SNP}$ = 0.74, s.e. = 0.03) and mean thickness (UKB $h^2_{SNP}$ = 0.61, s.e. = 0.02; ABCD $h^2_{SNP}$ = 0.78, s.e. = 0.02) showed the highest $h^2_{SNP}$ across both cohorts. LDSC[27] $h^2_{SNP}$ estimates from the meta-analysis ranged between 0.10 (s.e. = 0.01) and 0.18 (s.e. = 0.05) for area, and 0.12 (s.e. = 0.01) and 0.16 (s.e. = 0.02) for thickness, with the area of the genu showing the highest, and area of the splenium showing the lowest $h^2_{SNP}$ estimates. As shown in Supplementary Data 5–8, all LDSC and GCTA $r_G$ estimates between meta-analyzed CC phenotypes were significant.

## Gene-mapping and gene-set enrichment analyses

Gene-based association analysis in MAGMA[45] identified 30 genes for the total area, and 34 genes for total mean thickness of the CC, with 5 genes overlapping between area and thickness (*IQCJ-SCHIP1, IQCJ, BPTF, PADI2, CHIC2*). The strongest association seen with area was *AC007382.1* and the strongest association with mean thickness was *HBEGF* (Fig. 2a). There were between 15 and 31 genes for area, and between 7 and 25 genes for thickness identified within regions of the CC. Notably, *IQCJ, IQCJ-SCHIP1*, and *STRN* overlapped for all parcellations of CC area. *AC007382.1* overlapped for four out of five parcellations, and *STRN* and *PARP10* overlapped for three out of five parcellations of CC thickness (Fig. 2b, Supplementary Data 1–4). Enrichment of SNP heritability in 53 functional categories for each trait was determined via LDSC[46]. The majority of enrichment and the strongest effects across parcellations of the CC were observed in categories related to gene regulation/transcription in chromatin (Fig. 3a, b).

Gene-set enrichment analyses were also completed in MAGMA (Fig. 3c). The strongest effects of significant gene sets included those involved in postsynaptic specialization for total CC area, including GO:009901 (postsynaptic specialization, intracellular component) and GO:009902 (postsynaptic density, intracellular component). A theme of signal transduction-related pathways was observed for the splenium area, including R-HSA-6785631 (*ERBB2 regulates cell motility*) and R-HSA-8857538 (*PTK6 promotes HIF1A stabilization*). Enrichment of the "*CARM1 and regulation of the estrogen receptor*" was found for the posterior body thickness and is implicated transcriptional regulation via histone modifications. Enrichment of GO:1904714 (*regulation of chaperone-mediated autophagy*) was found for the isthmus area, which is implicated in lysosomal-mediated protein degradation. All significant results across all CC phenotypes are in Supplementary Data 18.

## Tissue-specific and cell-type-specific expression of corpus callosum associated genes

Gene-property enrichment analyses were completed in MAGMA with 54 tissue types from GTEx v8 and BrainSpan[47,48], which includes 29 samples from individuals representing 29 different ages, as well as 11 general developmental stages. An enrichment of genes associated with isthmus thickness were expressed in the cerebellum ($p_{(Bon)} = 0.017$). Area and thickness across parcellations of the CC showed an enrichment of expression of genes in the brain from early prenatal to late mid-prenatal developmental stages. An enrichment of expression of genes associated with area and thickness of the anterior body of the CC was observed in brain tissue prenatally, 9–24 weeks post conception. Enrichment of expression of genes associated with area of the genu was observed in brain tissue 19 weeks post conception. Enrichment of expression of genes associated with the total mean thickness of the CC was observed in brain tissue 19 weeks post conception. All results are

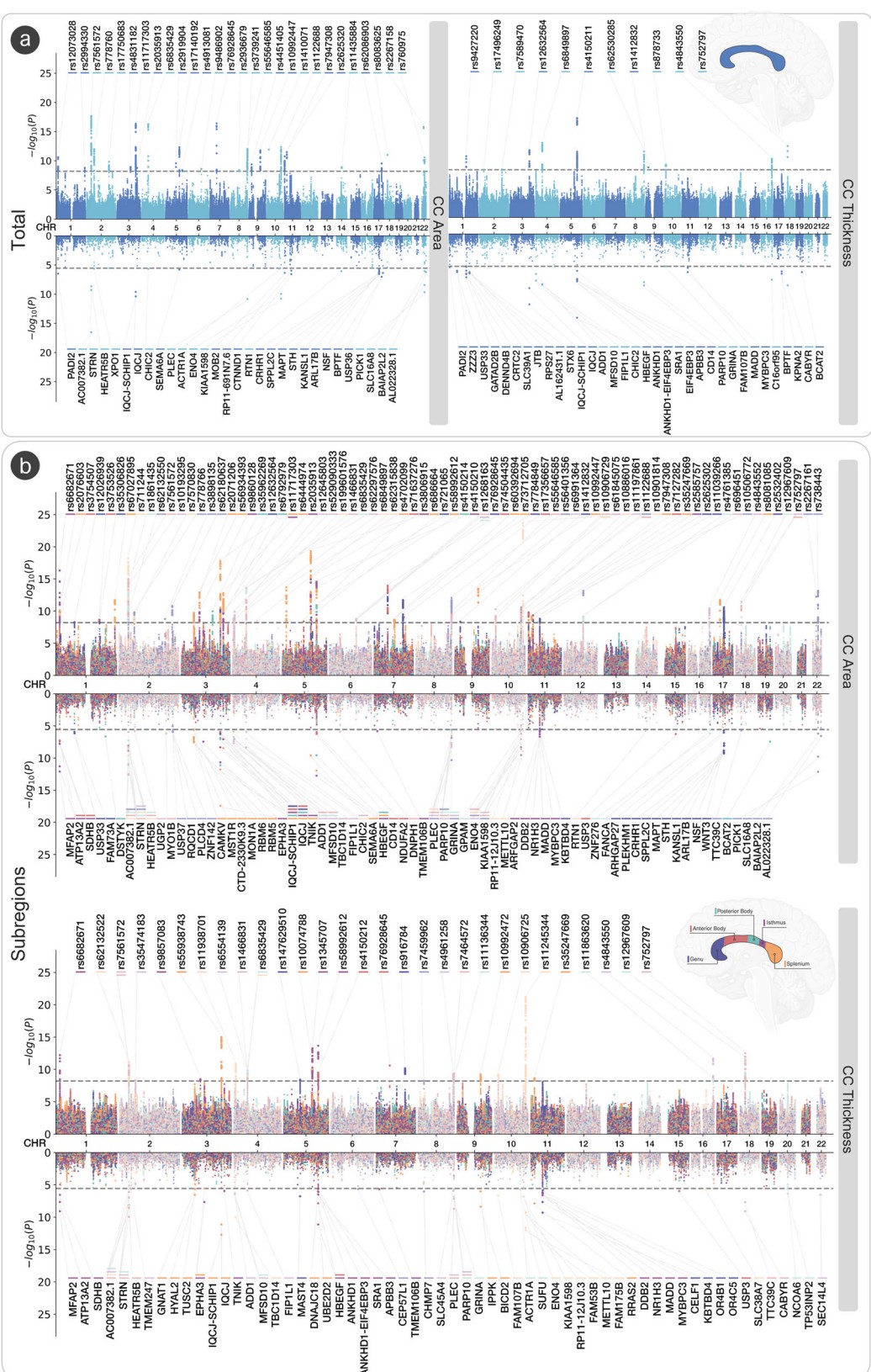

**Fig. 2 | GWAS meta-analysis of midsagittal corpus callosum area and thickness.**
**a** Miami plot for SNPs (*top*) and genes (*bottom*) based on MAGMA gene analysis for total area and total mean thickness. **b** Miami plot for SNPs (*top*) and genes (*bottom*) based on MAGMA gene analysis for area of thickness of the CC split by the Witelson parcellation scheme[23]. Results shown on the upper panels of (**a**) and (**b**) are from an inverse-weighted random-effects meta-analysis (DerSimonian-Laird method). Reported -log10(p-values) are two-sided. All loci are significant at the Bonferroni corrected, experiment-wide threshold of $p < 6.13 \times 10^{-9}$. Results shown on the lower panels of (**a**) and (**b**) are from the MAGMA gene-based analysis. Reported -log10(p-values) are two sided from the Z-statistic. All significant genes are shown at the Bonferroni corrected threshold of $p < 2.74 \times 10^{-6}$. Significant SNPs and genes are color-coded by CC traits. Created in part by using Biorender.com (agreement number PT28RS3SIJ).

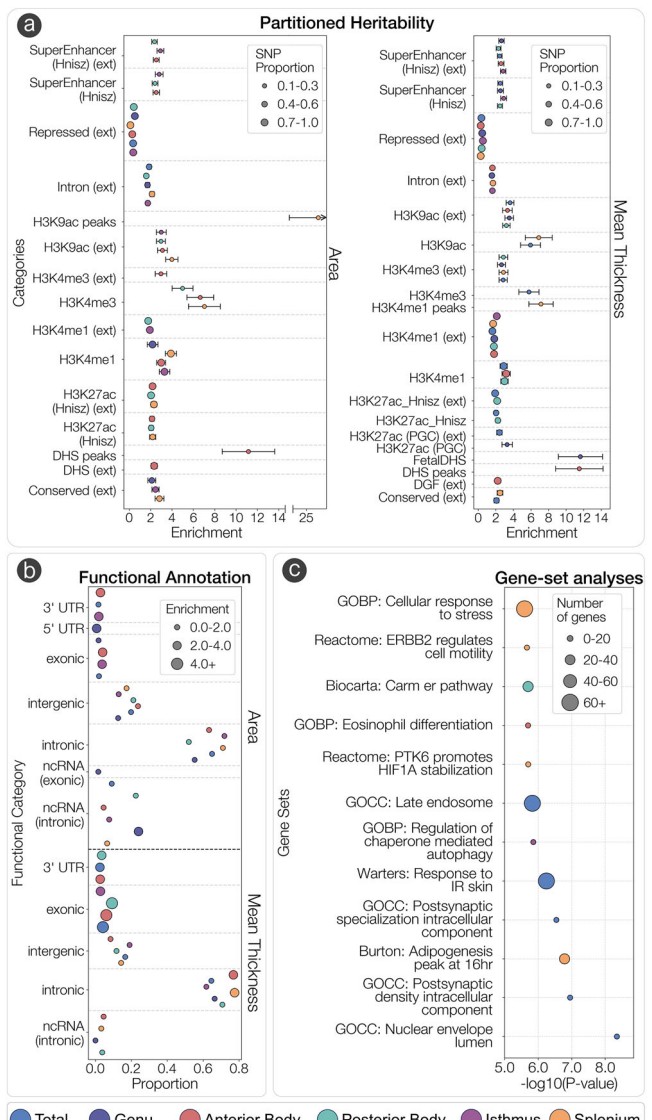

**Fig. 3 | Partitioned heritability, functional annotation and enrichment of gene-sets of CC morphology associated genetic variants. a** Significant enrichment of SNP heritability across 53 functional categories computed by LD Score regression for area (*left*) and mean thickness (*right*). Analyses were completed using the meta-analyzed GWAS summary statistics (*N* = 46,485). Data are presented as mean values +/− *s.e.* **b** Proportion of GWAS SNPs in each functional category from ANNOVAR across each CC phenotype. **c** Significant gene-sets across CC phenotypes computed via MAGMA gene-set analysis using the equivalent of a one-sided two-sample t-test at the Bonferroni corrected threshold of $3.23 \times 10^{-6}$. GOBP Gene-ontology biological processes, GOCC Gene-Ontology Cellular Components.

shown in Supplementary Data 19–21. These results, along with the gene-sets involved in histone modifications, were supported by LDSC-SEG analyses using chromatin-based annotations from narrow peaks[49], which showed a significant enrichment in the heritability by variants located in genes specifically expressed in DNase in the female fetal brain for total CC thickness ($p_{(Bon)}$ = 0.0105). Chromatin annotations showed a consistent and significant enrichment of splenium area and thickness-associated variants in histone marks of the fetal brain and neurospheres (Supplementary Data 25).

Using microarray data from 292 immune cell types, the area of the posterior body showed a significant enrichment in the heritability by variants located in genes specifically expressed in multiple types of myeloid cells ($p_{(Bon)}$ < 0.05), and the area of the isthmus showed

enrichment in innate lymphocytes ($p_{(Bon)}$ = 0.047). This further validates the aforementioned significant locus on gene *FOXO3*, which overlapped between the posterior body and isthmus (Supplementary Data 26).

Cell-type-specific analyses were performed in FUMA using data from 13 single-cell RNA sequencing datasets from the human brain. This tests the relationship between cell-specific gene expression profiles and phenotype-gene associations[50]. Of the 12 phenotypes tested, only total CC thickness showed significant results after going through the 3-step process using conditional analyses to avoid bias from batch effects from multiple scRNA-seq datasets. The most significant association was seen with oligodendrocytes located in the middle temporal gyrus (MTG, $p_{(Bon)}$ = 0.001) from the Allen Human Brain Atlas (AHBA). Oligodendrocytes ($p_{(Bon)}$ = 0.03) and non-neuronal cells ($p_{(Bon)}$ = 0.03) located in the lateral geniculate nucleus (LGN) from the AHBA also showed significant associations but were collinear (Supplementary Data 22).

LAVA-TWAS analyses[28,51] (Fig. 4) of expression quantitative trait loci (eQTLs) and splicing quantitative trait loci (sQTLs) of protein-coding genes in 16 different brain, cell type, and whole blood tissues revealed the strongest eQTL associations of area and thickness with *CROCC* expression in whole blood for the isthmus ($\rho = -0.53$, $p = 1.29 \times 10^{-10}$). Other notable eQTL (Supplementary Data 29) findings included total CC area and isthmus area and thickness being positively associated with *ATP13A2* expression in fibroblasts ($\rho = 0.48$, $p = 1.58 \times 10^{-7}$). The strongest sQTL association was a positive association observed with *KANSL1* (cluster 11710) in fibroblasts for genu area ($\rho = 0.83$, $p = 1.46 \times 10^{-14}$), which was the tissue type where most observed associations occurred across CC phenotypes (Supplementary Data 30). Moreover, a negative association was observed in a *KANSL1* (cluster 11707) in fibroblasts for the genu area ($\rho = 0.82$, $p = 3.11 \times 10^{-7}$). An sQTL in *MFSD13A* (cluster 7894) in the anterior cingulate showed very strong yet opposite associations for total CC thickness ($\rho = 0.42$, $p = 1.12 \times 10^{-13}$) and total CC area ($\rho = -0.44$, $p = 2.98 \times 10^{-11}$). Other notable findings across tissue types included *CRHR1* in the cortex, nucleus accumbens, and putamen, as well as *UGP2* in fibroblasts, whole blood, and the putamen. No significant results from LAVA-TWAS gene-set enrichment analyses were observed after Bonferroni correction (Supplementary Data 31–32).

## Genetic overlap of corpus callosum and cerebral cortex architecture

Broadly, we observed a pattern of negative genetic correlations with area and thickness of the CC with cortical thickness across regions of the cingulate cortex, but positive genetic correlations with regions' cortical thickness across the neocortex (Fig. 5a). Specifically, we observed a significant negative genetic correlation between total area with cortical thickness of the rostral anterior cingulate ($r_g = -0.35$, *s.e.* = 0.06) and posterior cingulate ($r_g = -0.28$, *s.e.* = 0.06). Mean thickness had a negative genetic correlation with cortical thickness of the rostral anterior cingulate ($r_g = -0.29$, *s.e.* = 0.06) and posterior cingulate ($r_g = -0.23$, *s.e.* = 0.05). Positive genetic correlations were observed with cortical thickness of the lingual gyrus ($r_g = 0.26$, *s.e.* = 0.05) and cuneus ($r_g = 0.27$, *s.e.* = 0.06). When parcellating by the Witelson scheme[23], negative genetic correlations were observed for area and mean thickness with cortical thickness of regions across the cortex and the cingulate, but positive genetic correlations with regions in the occipital lobe. We also observed a significant negative genetic correlation between total area of the CC with surface area of the precuneus ($r_g = -0.20$, *s.e.* = 0.04). (Supplementary Data 9–10).

Genetic correlations can reflect direct causation, pleiotropy, or genetic mediation. To explore potential causal relationships between CC phenotypes and morphometry of the cerebral cortex, we ran Generalized Summary-data-based Mendelian Randomization (GSMR)

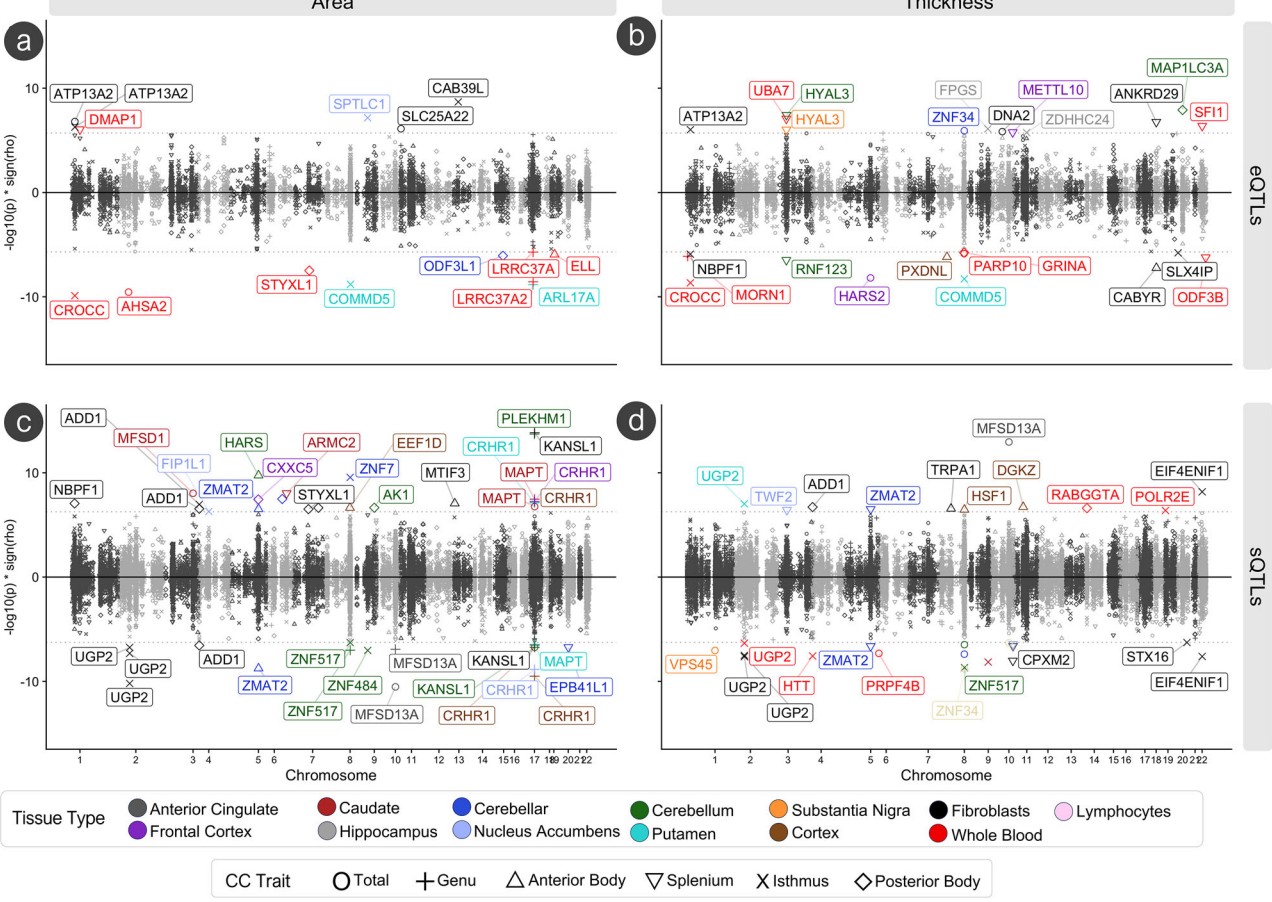

**Fig. 4 | LAVA-TWAS analyses of corpus callosum traits with gene-expression (eQTLs) and splicing (sQTLs).** Results of local genetic correlations between CC traits and eQTLs and sQTLs from GTEx v8 using the LAVA-TWAS framework. All significant points are colored by tissue type and labeled by CC trait. Significance was tested as a two-sided t-test statistic. Significance thresholds for eQTLs ($p < 2.01 \times 10^{-6}$) and sQTLs ($p < 5.45 \times 10^{-7}$) were determined by Bonferroni correction. Associations between (**a**) CC area and eQTLs, (**b**) CC thickness and eQTLs, (**c**) CC area and sQTLs, and (**d**) CC thickness and sQTLs are shown via -log10p values scaled by the direction of association (y-axis) and chromosomal location (x-axis).

analyses[52] directional effect of CC phenotypes on morphometry of the cerebral cortex, but not vice-versa. (Fig. 5b, Supplementary Data 14). There was a strong negative unidirectional effect of total CC area on the precuneus surface area ($b_{xy} = -0.50$, s.e. = 0.13, $p = 0.0002$), implying a greater total area and thickness of the CC results in a lower surface area of the precuneus. There was also a negative unidirectional effect of total CC mean thickness and cortical thickness of the posterior cingulate ($b_{xy} = -0.02$, s.e. = 0.008, $p = 0.02$), but not vice versa. When using the Witelson parcellation scheme, there was a strong negative unidirectional effect on the area of the genu on the cortical thickness of the rostral anterior cingulate ($b_{xy} = -0.001$, s.e. = 0.0003, $p = 0.003$).

Local genetic correlations of area phenotypes of the CC and surface area of the cerebral cortex with LAVA[28] showed many significant negative correlations in genes between the total area and posterior body and the precuneus SA along the 2p22.2 cytogenetic band (*QPCT, PRKD3, SULT6B1, NDUFAF7, EIF2AK2, HEATRSB, GPATCH11, CEBPZ, CEBPZOS, CDC42EP3, STRN, VIT*) (Fig. 5c, d). Negative genetic correlations between total CC area and caudal middle frontal gyrus SA in 5 genes along the 17q24.2 cytogenetic band (*HELZ, PSMD12, PITPNC1, ARSG, BPTF*) were also observed. Positive local genetic correlations along the 2p22.2 cytogenetic band were observed with the anterior body area and the surface area of the posterior cingulate (*CDC42EP3, PRKD3*), as well as the total area of the CC and precentral gyrus surface area (*HEATRSB*).

Many negative local genetic correlations were observed with mean thickness of the splenium and cortical thickness of the superior parietal gyrus (*TEX36, EDRF1, UROS, BCCIP, DHX32*) and the parahippocampal gyrus (*ZNF879*) along the 10q26.13–10q26.2 cytogenetic bands, while positive genetic correlations were observed with isthmus cingulate cortical thickness along the 10q26.13–10q26.2 cytogenetic bands (*EDRF1, TEX36, UROS, BCCIP, DHX32, CTBP2, CPXM2, GPR26, ZRANB1, FAM53B*).

The area of the posterior body showed a negative local genetic correlation with the cortical thickness of the pericalcarine gyrus (*GPATCH11*). The area of the isthmus showed positive local genetic correlations with the cortical thickness of the superior parietal gyrus (*LRRC73*), caudal middle frontal gyrus (*GPATCH2L*), and isthmus cingulate (*PLPPR3, CFD, R3HDM4, PTBP1, ELANE, MED16, PALM*) along the 19p13.3 cytogenetic band.

The mean thickness of the posterior body showed negative local genetic correlations with the surface area of the lingual gyrus (*STC2, NKX2-5*, 5q35.2) and pericalcarine gyrus (*NKX2-5*). Mean thickness of the isthmus showed negative local genetic correlations with the precuneus (*EIF2AK2, GPATCH11*, 2p22.2) and superior frontal gyrus (*TBX19*) surface area. Total mean thickness of the CC showed a positive genetic correlation with the surface area of the insula (*PDZRN3*). The mean thickness of the anterior body showed positive local genetic correlations with surface area of the superior parietal gyrus (*RETN, FCER2*). Splenium mean thickness showed positive genetic correlations

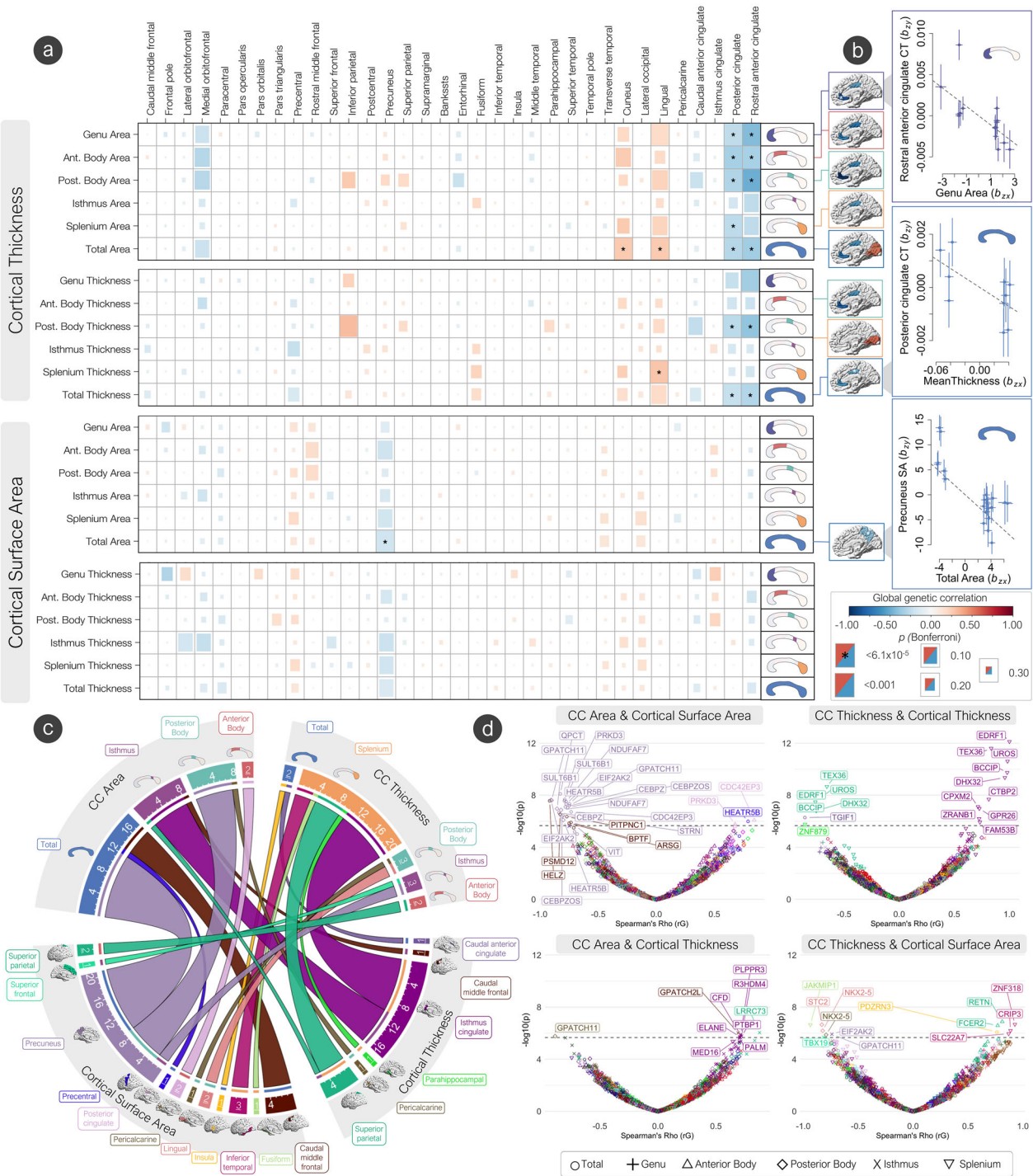

**Fig. 5 | The genetic overlap of the corpus callosum and cerebral cortex. a** Global genetic correlations (LDSC - rG) between CC phenotypes and cerebral cortex phenotypes. Significance was based on a two-sided Z-statistic. The Bonferroni significance threshold was set at $p = 6.1 \times 10^{-5}$. Surface area and cortical thickness of significant cortical regions with each CC phenotype are displayed on brain plots. **b** Of the significant global genetic correlations, significant Mendelian randomization (GSMR) results are displayed, representing the effect of CC phenotypes on cortical phenotypes free of non-genetic confounders. Significance was determined using the two-sided t-statistic calculated within GSMR. The number of SNPs used in GSMR were N = 26, N = 18, and N = 10 for the precuneus, rostral anterior cingulate,

and posterior cingulate respectively. Data are presented as beta values +/- *s.e.* **c** Chord plot displaying the number of significant bivariate local genetic correlations (LAVA) between CC and cortical phenotypes. Underlined numbers represent the total number of genes shared with that phenotype. **d** Volcano plots showing degree (-log₁₀ *p*-values) and direction (rG) of local genetic correlations (LAVA) between cortical and CC phenotypes. Significance was tested as a two-sided t-test statistic. Colors represent cortical regions labeled on the chord plot in section C. Significant genes (Bonferroni significance threshold was set at $p = 2.18 \times 10^{-6}$) across all phenotypes are labeled.

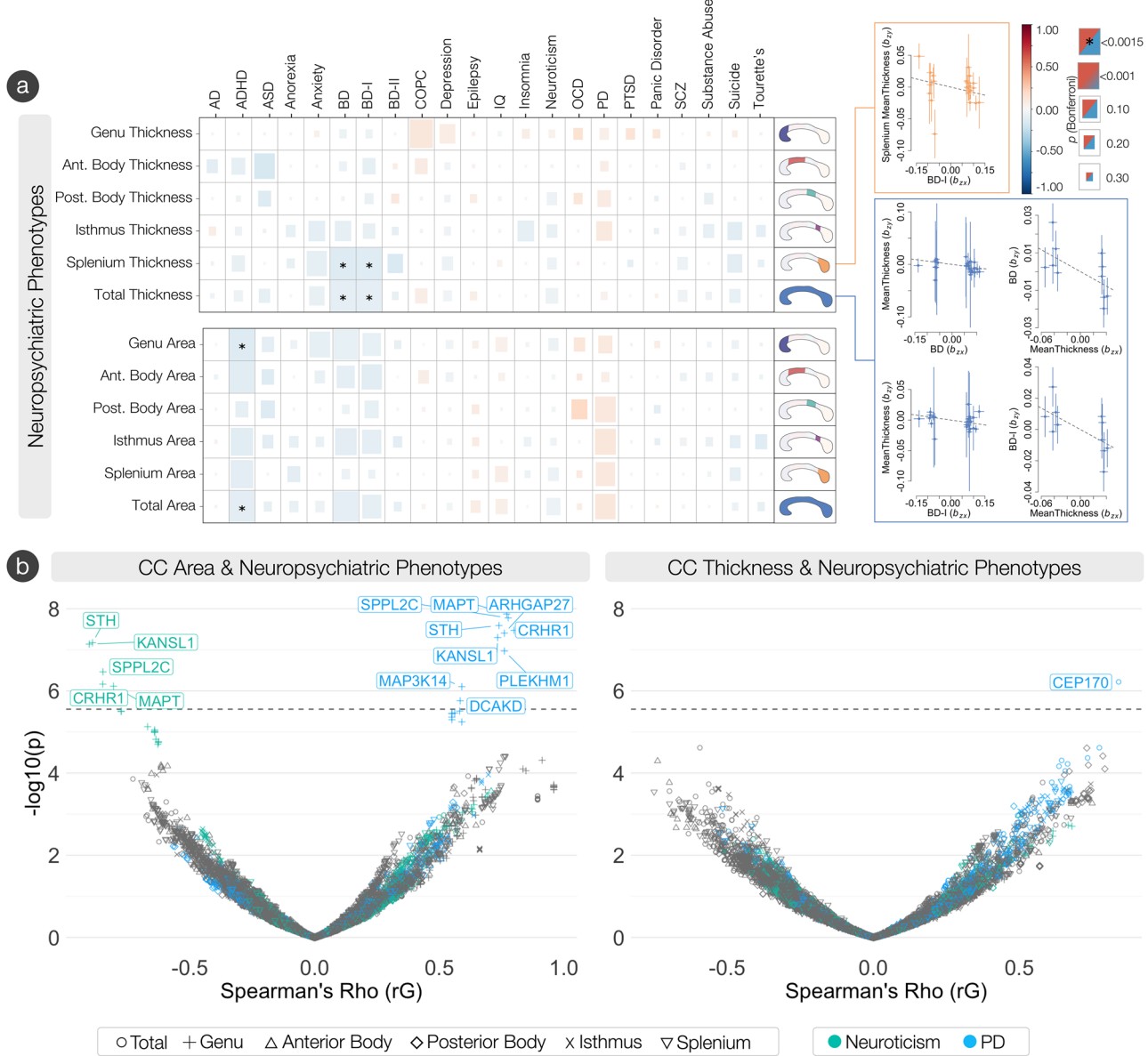

**Fig. 6 | The genetic overlap of the corpus callosum and neuropsychiatric phenotypes. a** Global genetic correlations between CC traits and neuropsychiatric phenotypes. Significance was based on a two-sided Z-statistic. Significant results are designated by the * at the Bonferroni significance threshold of $p = 0.0015$. Significant negative genetic correlations are observed between total and splenium thickness, and bipolar disorder (I). Significant negative genetic correlations are also observed with CC area phenotypes and ADHD. Of the significant global genetic correlations, significant Mendelian randomization (GSMR) results are displayed, representing the effect of CC phenotypes on neuropsychiatric phenotypes free of non-genetic confounders. The number of SNPs used in the GSMR analysis were N = 29 for BD on total mean thickness, N = 26 for BD-I on total mean thickness, N = 11 for total mean thickness on BD, N = 25 on BD-I on splenium mean thickness, and N = 11 on total mean thickness on BD-I. Data are presented as beta values +/− *s.e.* **b** Volcano plots showing degree (-log$_{10}$ *p*-values) and direction ($r_G$) of local genetic correlations (LAVA) between neuropsychiatric and CC phenotypes. Significant local

negative genetic correlations on the *STH, KANSL1, SPPL2C, CRHR1, and MAPT* genes are observed between the genu area and neuroticism. Significant local positive genetic correlations on the *MAPT, SPPL2C, STH, CRHR1, ARHGAP27, KANSL1, PLEKHM1, MAP3K14, and DCAKD* genes are observed between the genu area and Parkinson's disease. A significant positive local genetic correlation is observed on the *CEP170* gene between total mean thickness and Parkinson's disease. Significance was tested as a two-sided t-test statistic. Phenotypes with significant associations are colored (IQ and bipolar II disorder). Significant genes (Bonferroni significance threshold was set at $p = 2.79 \times 10^{-6}$) across all neuropsychiatric phenotypes are shown. AD Alzheimer's disease, ADHD attention deficit hyperactivity disorder, ASD autism spectrum disorder, BD bipolar disorder, BD-I bipolar I disorder, BD-II bipolar II disorder, COPC chronic overlapping pain conditions, IQ intelligence quotient, OCD obsessive-compulsive disorder, PD Parkinson's disease, PTSD post-traumatic stress disorder, SCZ schizophrenia.

with inferior temporal gyrus surface area (*ZNF318, CRIP3, SLC22A7*) along the 6p21.1 cytogenetic band.

### Genetic overlap of corpus callosum and associated neuropsychiatric phenotypes
We observed a significant genetic correlation (Fig. 6a, Supplementary Data 11) between total CC area and ADHD ($r_g = -0.11$, *s.e.* = 0.03),

bipolar disorder (BD, $r_g = -0.10$, *s.e.* = 0.03), and bipolar I disorder (BD-I, $r_g = -0.10$, *s.e.* = 0.03). Total mean thickness was genetically correlated with BD ($r_g = -0.10$, *s.e.* = 0.03) and BD-I ($r_g = -0.10$, *s.e.* = 0.03). When analyzing the regional Witelson parcellations[23], the area of the genu was genetically correlated with ADHD risk ($r_g = -0.13$, *s.e.* = 0.03), and the mean thickness of the splenium was genetically correlated with risk for BD ($r_g = -0.13$, *s.e.* = 0.03) and BD-I ($r_g = -0.12$, *s.e.* = 0.03).

GSMR analyses showed causal bidirectionality of genetic liability of BD ($b_{xy}$ = −0.06, $s.e.$ = 0.02, $p$ = 0.006) and BD-I ($b_{xy}$ = −0.05, $s.e.$ = 0.02, $p$ = 0.003) on total mean thickness of the CC, and mean thickness of the CC on BD ($b_{xy}$ = −0.19, $s.e.$ = 0.08, $p$ = 0.01) and BD-I ($b_{xy}$ = −0.23, $s.e.$ = 0.09, $p$ = 0.02). When using the Witelson parcellation[23], GSMR analyses showed causal directionality of genetic liability of BD-I on mean thickness of the splenium ($b_{xy}$ = −0.09, $s.e.$ = 0.04, $p$ = 0.01), but not vice versa (Fig. 6a, Supplementary Data 15).

Local genetic correlations with LAVA[28] (Fig. 6b, Supplementary Data 17) showed 5 negative local genetic correlations between area of the genu and neuroticism on the 17q21.31 cytogenetic band (*STH, KANSL1, SPPL2C, CRHR1, and MAPT*), and 9 positive local genetic correlations between anterior body area and Parkinson's disease (PD) on the 17q21.31 cytogenetic band (*MAPT, SPPL2C, STH, CRHR1, ARHGAP27, KANSL1, PLEKHM1, MAP3K14 and DCAKD*). One positive local genetic correlation was also observed between total mean thickness of the CC and PD (*CEP170*).

## Discussion

We conducted a GWAS meta-analysis of CC area and thickness across two cohorts with vastly different age ranges, leveraging our artificial intelligence-based tool, *SMACC*, to extract detailed CC phenotypes from 46,685 individuals across the UKB and ABCD studies. While prior research into the genetic basis of CC structure and development has primarily relied on candidate gene approaches in animal models and post-mortem human studies, our work addresses the notable differences between the human CC and its counterparts in animal models[6]. This study offers genome-wide insights into the genetic architecture of the human CC in vivo, significantly advancing our understanding of its development and variation. Previous GWAS efforts focused on total and parcellated CC volume using FreeSurfer-derived measures[53] solely in the UKB cohort[16,17]. Here, we expand on the well-replicated finding that area and thickness measures of neuroimaging phenotypes have distinct genetic influences[25]. Our findings strongly support this distinction, as our meta-analysis revealed zero overlapping significant loci between the area and thickness phenotypes of the CC. This underscores the value of separating these metrics to identify unique genetic contributions to CC morphometry. Furthermore, while previous studies have reported genetic correlations between CC FA and volume with neuropsychiatric traits such as bipolar disorder and ADHD[16,17], our investigation extends these findings by exploring the specific genetic influences on CC area and thickness. This approach enables a deeper understanding of the mechanistic underpinnings behind these associations. Notably, we identified localized, distinct genetic relationships between CC morphometry and traits like neuroticism and PD - associations that had not been previously reported.

To better understand the stability of genetic influences on CC morphometry across development, we estimated the genetic correlation of each trait between the UKB (adult) and ABCD (adolescent) cohorts. While initial LDSC-based estimates yielded low/imprecise genetic correlations - likely due to low sample size in ABCD - we complemented this with bivariate GREML using individual-level data via GCTA[40]. GCTA provides more precise estimates by modeling genetic covariance between traits directly and is more robust to smaller sample sizes[54]. We observed consistent and significant genetic correlations across all CC traits ($r_g$ ≈ 0.40–0.49, Supplementary Data 6), suggesting a partially shared genetic architecture of the CC across the lifespan. While some genetic effects are stable from childhood into adulthood, others may be population specific. The differences could reflect changes in gene expression, shifts in variant effect sizes, or age-dependent neurobiological processes (e.g., neural migration in early life vs. myelination in adulthood)[55]. Environmental and cohort-specific influences are likely to modulate these effects. Future GWASes using a larger sample size of individuals across the lifespan will be crucial to dissect developmental effects and genetic pathways of CC morphometry.

We show that the genetic architecture of the CC is highly polygenic, and specific genetic variants influence CC subregions along a rostral-caudal gradient. Five loci that were positionally mapped to genes were identified to influence both total area and mean thickness of the CC (*IQCJ-SHIP1, FIP1L1, HBEGF, CDKN2B-AS1*, and *FAM107B*). *IQCJ-SHIP1* had the strongest effect across total area and mean thickness, implicating mechanisms such as conduction of action potentials in myelinated cells via organizing molecular complexes at the nodes of Ranvier and axon initial segments, calcium-mediated responses, as well as axon outgrowth and guidance[56]. The strongest locus for total area was mapped to the *STRN* gene. *STRN* has been heavily implicated in the Wnt signaling pathway, which controls the expression of genes that are essential for cell proliferation, survival, differentiation, and migration via transcription factors[57–59]. The *HBEGF* gene was the strongest locus for total mean thickness, implicating mechanisms in early development. *HBEGF* expression is localized in the ventricular zone and cortical layers during development[60], and has been implicated in regulating cell migration via chemoattractive mechanisms[60]. Significant enrichment of heritability of total mean thickness in various histone marks from chromatin data (ATAC-seq) of the fetal brain and cortex derived primary cultured neurospheres, significant tissue expression in the brain 19-weeks post conception, as well as enrichment of gene sets involving regulation of histone modification, suggests genetic variants in regions of open chromatin and transcriptional activity regulation in early development are key mechanisms underlying CC morphometry. When histones are acetylated, they become more negatively charged. This negative charge repels the negatively charged DNA, causing the DNA to be "pushed away" from the histones. This loosening of the DNA-histone complex makes it easier for transcription factors to access the DNA and initiate transcription[61].

Parcellation of the CC into the five regions defined by the Witelson scheme allowed for further refinement and genetic understanding of its morphometry in a rostral-caudal gradient. Our results provide insight as to which molecular mechanisms influence this functionally defined gradient (i.e., prefrontal, premotor/supplementary motor, primary motor, primary sensory, and parietal/temporal/occipital)[24]. An overlap of genetic loci along the most anterior (genu and anterior body, *SHTN1*) and most posterior (isthmus and splenium, *IQCJ-SCHIP1*) regions of the CC, along with splenium heritability enrichment of histone chromatin marks of the fetal brain and dorsolateral prefrontal cortex, implicates regulation of neuron migration and action potential conduction. But the overlap of the *FOXO3* along the area of the posterior body and isthmus implicates IL-9 signaling and *FOXO*-mediated transcription responsible for triggering apoptosis[62]. Only the posterior body and isthmus showed heritability enrichment in immune cells, including myeloid cells and innate lymphocytes. The thinning of the CC (along the posterior body and isthmus) occurs in a functional gradient connecting the somatosensory and parietal association areas of the brain[6,63,64]. Such thinning of the posterior body and isthmus aligns with activity dependent pruning by functional area[6], where somatosensory circuits are pruned in early development in an experience dependent context[65]. As immune cells are increasingly being recognized as key players in brain maturation and neurodevelopment[66], our results suggest IL-9 mediating a neuroprotective effect in the CC during the cell dieback phase[66,67], and may play a significant role in posterior CC morphometry. LAVA-TWAS results showed another potential mechanism of isthmus pruning via expression of *ATP13A2* in fibroblasts, and splicing of genes involved in NF-κB signaling[68]. *ATP13A2* is involved in lysosomal-mediated apoptosis[69], suggesting such regulation of fibroblast mediated growth of callosal projections[70]. This hypothesis is also supported by the current discovery of enrichment of genes related to isthmus area in the

"*regulation of chaperone mediated autophagy pathway*", which may influence isthmus morphometry.

The topographic organization of the CC correlates with the homotopic bilateral regions of the cortex it is known to connect[5]. A variety of genetically regulated principal mechanisms influence CC neuronal and glial proliferation, neuronal migration and specification, midline patterning, axonal growth and guidance, and post-guidance refinement to homotopic analogs in the cortex[71,72]. Our results suggest potential genetic mechanisms contributing to callosal-cortical organization. We show an overall negative global genetic correlation of CC phenotypes with the cortical thickness of the cingulate and surface area of the posterior parietal cortices, including a unidirectional negative effect of genu area on rostral anterior cingulate thickness, and total area on precuneus surface area free of any non-genetic confounders. Positive global genetic correlations of total CC area and splenium thickness with cortical thickness in the occipital cortex were also observed. Local genetic correlations of the CC were observed throughout the cerebral cortex, most pronounced with total CC area and splenium thickness. Notable findings included numerous genes in the chr2p22 cytogenetic band showing negative correlations between total CC and posterior body area with precuneus surface area, including the significant *STRN* gene observed across all CC phenotypes, further implicating the Wnt signaling pathway and dendritic calcium signaling in the context of neurodevelopment[73,74]. Within this cytogenetic band, *HEATR5B* was also positively genetically associated with precentral gyrus surface area. Opposing genetic effects were observed between splenium thickness with isthmus cingulate thickness (i.e. positive) vs. superior parietal cortex thickness (i.e. negative) in genes in the chr10q26.13 cytogenetic band. Previous clinical neuropsychiatric conditions associated with copy number variations of chr10q26.13 include abnormal cranium development, global developmental delay and learning difficulties, and neurodevelopmental manifestations including ADHD or autistic behaviors[75–77]. This finding provides a novel testable hypothesis for functional follow up studies, as alterations in the isthmus cingulate and superior parietal cortex have been observed in large-scale studies of various neurodevelopmental disorders[78]. Positive genetic associations in the chr19p13.3 cytogenetic band were observed between the isthmus area and isthmus cingulate cortical thickness, which has been implicated with microcephaly, ventriculomegaly and developmental delay[79,80].

Our results demonstrate opposing genetic relationships between CC phenotypes (area and thickness of the entire CC and it's sub-regions) and thickness of the cingulate cortex (negative) vs the neocortex (positive), which suggests a strong genetic component underlying the development of the CC via pioneer axons and chemotaxis. Coupled with the observed negative phenotypic correlations (Supplementary Data 13), this suggests that the relationship between the CC and the thickness of the cingulate cortex (but not surface area) is influenced by distinct genetic mechanisms that govern their development. Developmentally, pioneer axons emerge in the cingulate and project their axons across the midline using guidance cues. A large portion of these callosal projections are pruned and myelinated in an activity-dependent manner, such that axonal remodeling is highly dependent on correlated neural activity in the cortex[6,81–83]. The strongest local genetic correlation supporting this finding was observed between the total mean thickness of the CC and the rostral anterior cingulate thickness on *TGIF1*. As *TGIF1* is implicated in holoprosencephaly (i.e. where the brain fails to develop two hemispheres), forebrain development via alterations in the Sonic Hedgehog (SHH) pathway, and disruption of axonal guidance via chemoattractive mechanisms[84,85], these results provide a potential genetic localization for functional follow-up. The isthmus cingulate, in relation to the isthmus and splenium, was the only cingulate region showing positive local genetic correlations, providing further evidence of distinct molecular mechanisms (e.g. immune-mediated apoptosis and regulation of callosal projections) compared to the rest of the CC underlying its structure and development.

Abnormalities of the CC have also been associated with various neurological/neuropsychiatric disorders[6]. This study demonstrates a significant negative genetic relationship between the CC and ADHD (utilizing the latest ADHD GWAS findings)[86], and also replicates previously observed negative genetic associations with bipolar disorder[16,17]. It is important to note, that prior studies focused on brain volume phenotypes, whereas the current study examines area and thickness, which are known to be influenced by different genetic factors[25]. The negative global genetic correlations observed in the CC area with ADHD and CC thickness with bipolar disorder indicate that the allelic differences resulting in a smaller CC area and thickness are partly shared with those resulting in a greater risk for ADHD and bipolar disorder, respectively. Further evidence of the negative genetic relationship between ADHD and CC area is provided by studies that show the CC is smaller in individuals with ADHD across various ages[87–89], suggesting that impaired inter-hemispheric communication between sensorimotor and attentional systems may contribute to symptoms of hyperactivity, impulsivity, and inattention. Our results also provide a credence to future studies investigating the genetic relationship between the CC and bipolar disorder, as differences in the CC in bipolar disorder have been well established[13,90,91]. Negative local genetic correlations on the 17q21.31 cytogenetic band between genu area and neuroticism implicated the closely located *CRHR1* and *KANSL1* genes, which were also highly significant genes observed with genu area and splicing QTLs (sQTLs) in various cortical and subcortical tissue types in the TWAS analysis. Neuroticism, a construct historically describing a cluster of negative emotions, thoughts, and behaviors under the umbrella of "negative affect," is increasingly recognized as a significant predictor of susceptibility to stress-related psychiatric disorders, including anxiety and depression[92]. *CRHR1* encodes the corticotropin-releasing hormone receptor 1, a receptor widely expressed in the cortex and central nervous system that mediates the effects of corticotropin-releasing factor (CRF). The CRF system plays a central role in orchestrating the body's stress response through the hypothalamic-pituitary-adrenal (HPA) axis and autonomic nervous system. Dysregulation of this system has been extensively linked to the pathophysiology of stress-related anxiety and mood disorders, with neuroticism often serving as a measurable intermediate phenotype[92]. *CRHR1* polymorphisms have been associated with differential responses to stress and heightened susceptibility to psychiatric disorders[92], potentially mediated via altered connectivity between the prefrontal cortices via the genu and altered RNA splicing of the *CRHR1* gene inside cortical tissue.

Moreover, all of the local genetic correlations on the 17q21.31 cytogenetic band observed between the genu area and neuroticism were also observed with PD, but in the *positive* direction, suggesting heightened risk. The strongest association was with the microtubule-associated protein tau (*MAPT*) gene. The same locus was also a highly significant sQTL with genu area in the caudate nucleus in the TWAS analysis. This sQTL falls within exon 7 of *MAPT*, part of the proline-rich domain of the tau protein[93]. This region is known to regulate microtubule binding and aggregation, with the 6p and 6d *MAPT* isoforms resulting in altered tau assembly and reduced aggregation[93,94]. Multiple independent studies have linked *MAPT* to PD risk[95–98], and our findings suggest that these splicing events may influence the structural morphology of the genu of the corpus callosum via effects on the caudate nucleus, a region heavily implicated in PD pathology[99,100]. The genu, which facilitates interhemispheric communication between the prefrontal cortices[6], shows longitudinal progressive structural decline in PD, with its degeneration correlating with worsening akinetic-rigid motor symptoms[101]. Lower genu volume has also been reported in PD patients, particularly those with cognitive impairment, compared to cognitively intact individuals[102]. The loss of dopaminergic neurons in

the substantia nigra and the broader nigrostriatal pathway, including within the caudate nucleus, is a hallmark of PD and is more pronounced in patients with mild cognitive impairment[103]. This finding suggests that altered splicing of *MAPT* in the caudate nucleus may contribute to increased risk for Parkinson's disease with cognitive impairment through shared genetic effects on the morphometry of the genu, a region critical for facilitating higher-order cognitive functioning.

Several factors can contribute to the lack of global and local genetic correlations of the CC with other tested traits. Our primary explanation is that the CC does not serve a core biological mechanism underlying the pathology of many of the tested traits. Methodological limitations in LDSC and LAVA may also account for a lack of significant results. LDSC requires GWAS summary statistics with a large sample size or a high chi-squared test-statistic. Discordant effects across the genome can also result in null findings. The LAVA workflow uses common SNPs among all the traits' GWAS summary statistics tested, which can reduce the number of variants used in the analysis. This reduction can decrease the power to detect heritability at each locus, and not meet the stringent multiple comparisons threshold in our analysis ($p < 0.05/18380 = 2.72 \times 10^{-6}$) for a bivariate correlation at each locus to be tested. Moreover, the strict Bonferroni threshold correcting for every test across every trait ($0.05/17909 = 2.79 \times 10^{-6}$) can reduce the likelihood of finding significant bivariate associations. Phenotypic variability and underlying genetic heterogeneity within the tested traits may also contribute to the absence of significant findings. For instance, complex conditions such as Alzheimer's disease[104,105] and chronic pain[106] are increasingly recognized as comprising multiple subtypes with distinct biological and clinical profiles. Consequently, stratified GWAS analyses, tailored to these subtypes, may be necessary to uncover genetic correlations with our specific neuroanatomical traits.

Functional enrichment analysis across CC area, thickness, and volume (from Chen et al.[17] and Campbell et al.[16] papers) revealed both shared and trait-specific biological signatures (Supplementary Data 40–41). Consistent across all modalities and subregions was strong enrichment of growth factor signaling, particularly PDGFRA/B-driven, PI3K/AKT-related pathways, and RAS-MAP kinase cascade pathways[107]. These pathways were significant using the results with and without ICV as a covariate. Growth factor receptor signaling pathways are important intracellular pathways for cellular survival, growth, and proliferation through various mechanisms, which require further study in the current context[108–110]. Area measures showed the greatest number of significant associations and the strongest enrichment of these pathways. However, thickness measures, especially in the isthmus, showed the strongest enrichment of these pathways across all morphometry measures and CC subregions. The thickness of isthmus was also the only measure which showed enrichment of the axonemal basal plate, which is crucial for the formation of cilia and flagella, ensuring proper motility of cells[111]. Volume measures, especially from the Campbell et al.[16] results, highlighted neuronal morphogenesis and guidance pathways - especially in the splenium. These findings suggest that while a core set of signaling pathways influence CC morphology broadly, distinct modalities may reflect different aspects of structural development and cellular regulation.

In summary, this work identifies genome-wide significant loci of morphometry of the overall CC and its sectors, convergence on biological functions with a particular importance of apoptosis and pruning during development, tissues and cell types, as well as the genetic overlap with the cerebral cortex and neuropsychiatric conditions.

## Methods

### Artificial intelligence corpus callosum extraction and segmentation with SMACC

We developed a UNet-based automated segmentation tool that segments mid CC in multiple modalities like T1w, T2, and FLAIR, assesses the quality of the segmentation using machine learning methods on the meaningful metrics extracted from the segmentation, and is generalizable to data from various scanners and sites. To our knowledge, there has been no published integrated pipelines for mid CC extraction with quality control in multiple MR modalities. Existing deep learning methods, like DeepnCCA, have been trained to segment mid CC but only work on T2w images and have been trained on data from one scanner only, so they might not be generalizable to data from other scanners. Other existing methods like FreeSurfer (which was used in previous CC GWAS studies)[16,17], FastSurfer and a few UNet based methods[112] segment CC but do not assess the quality of the segmentations.

**Data preprocessing.** All UKB participants completed a 31-min neuroimaging protocol using a Siemens Skyra 3 Tesla scanner and a 32-channel head coil in one of three MRI scanning locations. All 3D structural T1-weighted brain scans were acquired using the following parameters: 3D MPRAGE, sagittal orientation, in-plane acceleration factor = 2, TI/TR = 880/2000 ms, voxel resolution = 1 × 1 × 1 mm, acquisition matrix = 208 × 256 × 256 mm. All scans were pre-scan normalized using an on-scanner bias correction filter. More details of the imaging protocols may be found in the following reference papers[113,114].

All ABCD participants completed a neuroimaging protocol in one of three scanner types at 21 different sites[115]. The Siemens Prisma had the following parameters for the T1-weighted scans: TI/TR = 1060/2500 ms, TE = 2.88 ms, voxel resolution = 1 × 1 × 1 mm, acquisition matrix = 176 × 256 × 256, flip angle = 8 degrees. The Philips Achieva Ingenia had a TI/TR = 1060/6.31 ms, voxel resolution = 1 × 1 × 1 mm, acquisition matrix = 225 × 256 × 256 mm and a flip angle = 8 degrees. The GE MR750 had a TI/TR = 1060/2500 ms, TE = 2 ms, voxel resolution = 1 × 1 × 1 mm, acquisition matrix = 208 × 256 × 256, and a flip angle = 8 degrees.

All T1w MRIs were registered to MNI152[116–118] 1 mm space with 6 degrees of freedom using FSL's *flirt*[119] command.

**SMACC development and UNet training.** Mid-sagittal T1w, T2w, and FLAIR images from UK Biobank[21], PING[120], HCP[121], and ADNI1[122] were used for training the UNet model for CC segmentation. Individual study scanner parameters can be found in their respective references. The demographic information for the datasets used to create the UNet model is shown in Supplementary Data 31. Augmentation of image data is a common procedure in deep learning to prevent model overfitting and improve model accuracy[123]. All the images were downsampled by a factor of 2, 3, 4 and 5 along the sagittal axis and then upsampled back to the original size using MRtrix's *mrgrid* command to include low resolution images in the training[124]. To include lower resolution T1w images resembling older or clinical data in training, all the images were harmonized using a fully unsupervised deep-learning framework based on a generative adversarial network (GAN)[125] to a subject from the ICBM dataset[117]. The original images in the training set already had 5–10 degrees rotation variation, so we rotated images in increments of 15 degrees to include more variety of head orientations. Black boxes were randomly added to the images to imitate partial agenesis cases. Supplementary Fig. 1 shows some T1w augmented images that were the input training images for the UNet model.

**UNet implementation.** A Tensorflow implementation of UNet[126] was trained on 80% of the images for 250 epochs until the difference between the intersection over union (IOU) after consecutive iterations was less than $1 \times 10^{-4}$. The U-Net architecture is structured with a contracting pathway and an expansive pathway. The contracting pathway repeatedly performs two 3 × 3 convolutions (without padding), with each convolution followed by a rectified linear unit (ReLU) and a 2 × 2 max pooling operation. At each stage in the expansive

pathway, the feature map is upsampled, followed by a $2 \times 2$ convolution, which reduces the feature channels by half. Then, the corresponding cropped feature map from the contracting pathway is concatenated, and two $3 \times 3$ convolutions are applied, with each one followed by a ReLU. We used the following training parameters: $1 \times 10^{-4}$ learning rate and an Adam optimizer[127]. The rest of the data was used for validation. The midsagittal CC (midCC) was initially segmented using image processing techniques[128] on subjects from ADNI1 (N = 1032, 54–91 years), PING (N = 1178, 3–21 years), HCP (N = 963, 22–37 years) and UKB (N = 190, 45–81 years). These masks were then visually verified and manually edited by neuroanatomical experts which served as the ground truth. To evaluate the model, the area of overlap between the predicted segmentation and the ground truth was calculated.

**CC shape metrics extracted with SMACC.** *SMACC* provides outputs of global and regional shape metrics extracted from the CC segmentation, including area, thickness, length, perimeter and curvature. The regional shape metrics were based on a 5 compartment version of the Witelson atlas[23,24]. The Witelson atlas is composed of the (1) genu, (2) anterior midbody, (3) posterior midbody, (4) isthmus, and (5) splenium. The metrics used for the GWAS analysis were area and mean thickness of the total CC and all of the parcellations of the Witelson atlas. The thickness is defined as the distance in the inferior-superior direction between the top and bottom of the contour and at every point along the length of the segment, then averaged across the region of interest. The total area is the summation of the number of voxels with intensity value greater than 0.5 in the segmentation.

**Corpus callosum segmentation quality control (QC) with SMACC.** To ensure that segmentations were of appropriate quality without having to manually assess all output images, which eventually may scale to hundreds of thousands of scans, we included an automated quality control (QC) assessment into *SMACC*. The regional and global metrics were used as inputs to the machine learning models detailed below for automatic binary classification of segmentations as Pass or Fail. CC segmentations from *SMACC* were manually assessed across multiple datasets by neuroanatomical experts. This included data from UKB (N = 12,902, aged 45–81 years), ADNI1 (N = 724, aged 54–91 years), PING (N = 857, 3–21 years) and HCP (N = 615, 22–37 years), all of which served as the ground truth for QC model building. All data was split 80/20 for training/testing.

Figure 7 gives the overview and the flow of *SMACC*. Several architectures, including a 3-layer sequential neural network with 42 neurons, 22 in the second layer, and 11 in the third layer; a wide & deep neural network with 80 neurons in the first 3 layers and 40 in the last 3 layers, an XGBoost classifier, and an ensemble model were tested to classify the segmentations from the UNet as pass or fail. The ensemble model consisted of XGBoost, k-nearest neighbors (KNN), support vector classifier (SVC), logistic regression, and a random forest classifier. The results from all the classifiers in the ensemble model were combined using a majority voting classifier. All the models were compared using metrics including precision, recall, F1 score, and Area Under the Curve (AUC). Supplementary Data 34 shows the performance of different models based on the shape metrics extracted from the CC segmentations.

**SMACC vs FreeSurfer**
**Comparing SMACC and FreeSurfer via Dice scores with respect to manual masks.** For assessing the accuracy of the *SMACC* compared to the ground truth and compared to the commonly used tool FreeSurfer[129], we ran the *SMACC* pipeline on 30 subjects from the Hangzhou Normal University (HNU) test-retest dataset[130,131]. Each subject in this dataset was scanned with a full brain T1w MRI 10 times within a period of 40 days, for a total of 300 scans. All 300 scans had

also been manually segmented by a neuroanatomical expert to serve as the ground truth. Segmentations from *SMACC* and FreeSurfer v7.1 were compared to manual segmentations using the Dice overlap coefficient. The average Dice coefficient between automated CC masks from *SMACC* and ground truth segmentations was 0.94 across all scans. The average Dice score between FreeSurfer CC segmentations and manual masks was 0.82. The Dice score was consistently higher for all the subjects using *SMACC*. Supplementary Fig. 2 and Supplementary Data 35, show a few midCC segmentations obtained from *SMACC* compared to FreeSurfer.

**ICC for SMACC.** To assess test-retest reliability of *SMACC* the intraclass correlation (ICC) scores were calculated. Average ICC values for thickness and area of the Witelson parcellations and the total CC were greater than 0.9 and are shown in Supplementary Fig. 3.

**Study cohorts**
**U.K. Biobank.** The UK Biobank (UKB) is a large population level cohort study conducting longitudinal deep phenotyping of around 500,000 participants in the United Kingdom (UK) aged between 40–69 at recruitment. All participants provided informed consent to participate. The North West Centre for Research Ethics Committee (11/NW/ 0382) granted ethics approval for the UK Biobank study[21]. We used genotype data from UKB released in May 2018. The data was collected from 489,212 individuals, and 488,377 of those individuals passed quality control checks by UKB. The genotypes were then imputed using two reference panels: the Haplotype Reference Consortium (HRC) reference panel and a combined reference panel of the UK10K and 1000 Genomes projects Phase 3 (1000G) panels[21]. There were 8,422,770 SNPs following quality control (QC) of the data which included having a genotyping call rate (SNPs missing in individuals) of greater than 95%, removing variants with a minor allele frequency less than 0.01 (1%), removing variants with Hardy-Weinberg equilibrium p-values less than $1 \times 10^{-6}$, and removing individuals with greater than three standard deviations away from the mean heterozygosity rate. To determine a conservative European ancestry in UKB, the ENIGMA MDS protocol (https://enigma.ini.usc.edu/protocols/genetics-protocols/) was completed using 10 components. The mean and standard deviations of the first and second genetic components of individuals who were classified as Utah residents with Northern and Western European ancestry from the CEPH collection (CEU) from the HapMap 3 release were then calculated. Individuals in UKB who were within a distance of 0.0101 on components 1 and 2 were classified as of European ancestry (Mean age: $64.73 \pm 7.86$ years, range 45.17–82.75; Sex: 21,717 females; N = 41,979). The MDS plot of individuals included in the analysis overlaid over the HapMap 3 population is available in Supplementary Fig. 4.

**ABCD.** The Adolescent Behavioral Cognitive Development (ABCD) study is the largest study in the United States (USA) following adolescent children starting from 9 years of age through adolescence with deep phenotyping including neuroimaging and genotyping using the Smokescreen™ Genotyping array consisting of over 300,000 SNPs[115,132,133]. Only neuroimaging from baseline were used. Following imputation using the ENIGMA protocol[134] with the European 1000 Genomes Phase 3 Version 5 reference panel, phased using Eagle version 2.3[135], and the QC process as described in the UKB cohort, a total of 5,683,360 SNPs were included from individuals of European ancestry (Mean age: $9.5 \pm 0.5$ years, range 8.0 - 11.0; Sex: 2086 females; N = 4706). To determine European ancestry in ABCD, the methods described for the UKB were completed. The MDS plot of individuals included in the analysis overlaid over the HapMap3 population is available in Supplementary Fig. 5.

We also analyzed non-European ancestry individuals to examine the generalization of the observed effects across ancestries. In order to

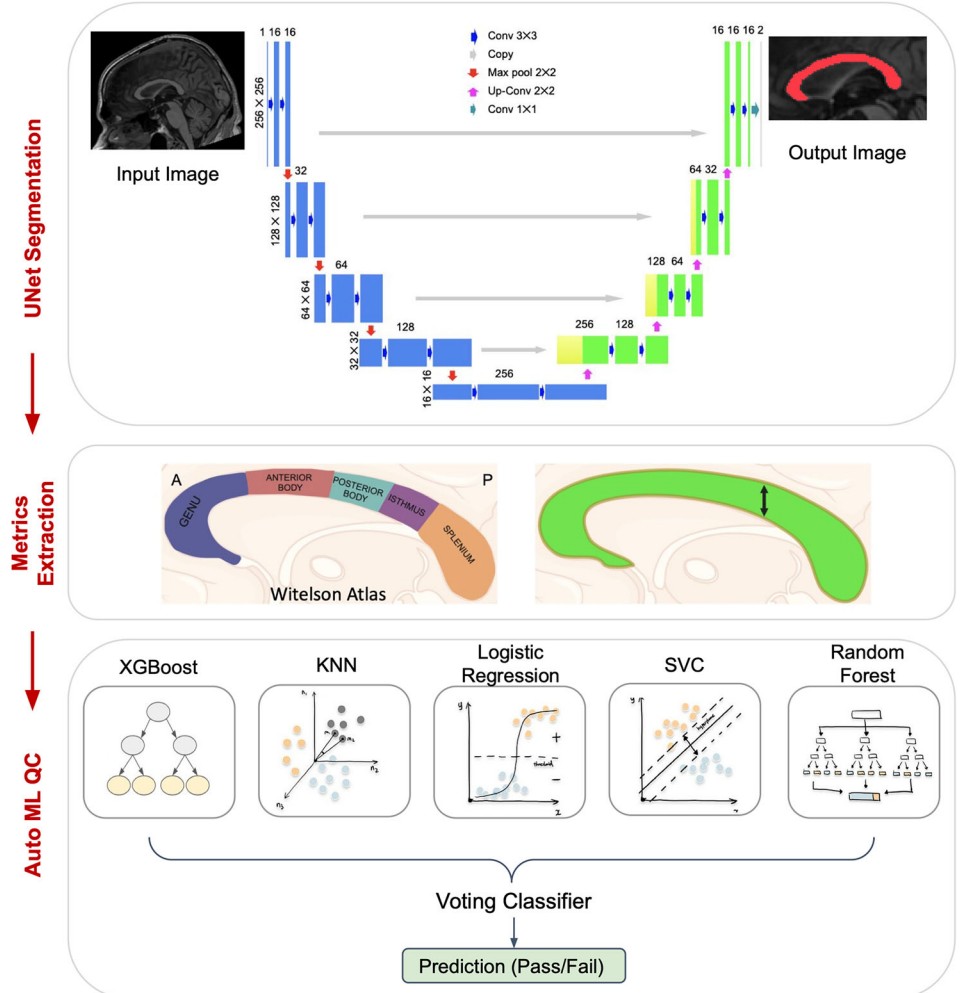

**Fig. 7 | Segment, Measure, and AutoQC the midsagittal CC (SMACC) pipeline.** The midsagittal slice from a participant registered to MNI space with 6 degrees of freedom serves as an input to the UNet architecture used for the midsagittal CC segmentation. The Witelson atlas was used for segmenting the CC into five different regions. Global and subregion metrics (thickness and area-shown in green) were extracted from the segmentation. The thickness (black arrow) is defined as the distance in the inferior-superior direction between the top and bottom of the contour, after reorientation to standard space, at every point along the length of the segment, then averaged across the region of interest. These metrics serve as input for the ensemble machine learning model used for labeling CC segmentations as having passed or failed quality control (QC). MNI Montreal Neurological Institute, CC corpus callosum, ML Machine Learning, KNN K Nearest Neighbors, SVC Support Vector Classifier.

accurately estimate non-European individuals using the HapMap3 reference panel, the KING software package[136], which uses a well-validated MDS and support vector machine approach, was used to estimate ancestry composition in all individuals not used in the principal GWAS (Supplementary Data 36). While making sure no individuals classified as CEU, TSI, or a combination of both, were not included, we included 636 individuals from the UKB (Mean age: 62.1 ± 8.3 years, range 44.6–80.2; Sex: 286 females) and 4129 individuals from ABCD (Mean age: 9.5 ± 0.5 years, range 8.0–11.0; Sex: 2009 females).

### GWAS meta-analysis of corpus callosum morphometry

Genome-wide association analysis (GWAS) for UKB and ABCD separately for all CC phenotypes were completed via a linear whole-genome ridge regression model using REGENIE, allowing for the control of genetic relatedness[137]. Covariates included age, sex, age*sex interaction, and the first 10 genetic principal components. A two-step REGENIE analysis was completed with the following parameters. For step 1, the entire dataset was used with a block size of 1000 and leave-one-out-chromosome validation[137]. Step 2 was completed with a threshold for minor allele count of 5, a block size of 1000, and otherwise default parameters.

A meta-analysis of GWAS summary statistics of all CC derived metrics in UKB and ABCD were conducted using METAL software and the random-metal extension[29,30], based on the random-effects model. A random-effects model was chosen since the effect sizes of SNPs on the CC has the potential to be different between the UKB and ABCD cohorts due to age. White matter volume is known to increase through childhood and start decreasing in middle adulthood[41], which may result in different genetic effect sizes being observed. We opted to conduct a meta-analysis instead of using a two-stage discovery-replication approach because Skol et al. have shown that this method is more powerful, despite using more stringent significance levels for multiple correction[138], and is common practice in the literature[86,98,139]. Percent variance ($R^2$) explained by each significant SNP was calculated using the approach described in Rietveld et al.[140]. The $R^2$ of each variant $j$ was calculated via:

$$R_j^2 \approx \frac{2p_j q_j \cdot \hat{\beta}_j^2}{\hat{\sigma}_y^2}$$

where $p_j$ and $q_j$ are the minor and major allele frequencies, $\hat{\beta}$ is the estimated effect of the variant within the meta-analysis and $\hat{\sigma}2$ is the estimated variance of the trait (for which we used the pooled variance of the trait across UKB and ABCD. In order to determine the number of independent traits, matrix spectral decomposition was computed using matSpD in R on the phenotypic correlations between CC traits using the method proposed by Li and Ji[141,142]. This resulted in 8.16 effective independent variables, and a significance threshold of $p = 5 \times 10^{-8}/8.16 = 6.13 \times 10^{-9}$. Meta-analyses were also completed for non-European individuals. To determine if the global measure of brain intracranial volume (ICV) would have an impact on the analysis, all GWAS were completed again using ICV as an additional covariate. All results comparing results without and with ICV as a covariate is shown in Supplementary Data 38.

To determine the degree of overlap of genes in the ICV vs no ICV analysis, the overlap coefficient, or Szymkiewicz–Simpson coefficient, was calculated[143]. In order to determine differences in enrichment in (1) gene ontology categories, (2) biological pathways, and (3) transcription factors, genes mapped to significant loci that were specific to the analysis with or without ICV as a covariate, or common to both, were separately entered into g:Profiler for a multi-gene list analysis[144]. All analyses were completed using the g:SCS threshold[145], all gene ontology categories, all biological pathway categories, and the TRANSFAC database.

## Heritability and genetic correlations within and between cohorts

To determine SNP heritability ($h^2_{SNP}$) tagged from SNPs used in the analysis, we used the GREML approach implemented in GCTA[43,44], while adjusting for the same covariates as in the GWAS. The SNP heritability ($h^2_{SNP}$) from LDSC[27], was also computed, which estimates heritability casually explained by common reference SNPs. Genetic correlations between the UKB and ABCD cohorts for area and thickness of each parcellation of the CC defined by the Witelson scheme, and total CC were completed using LDSC[27]. Between cohort heterogeneity of $h^2_{SNP}$ should not be considered unusual, as the genetic influence observed on the CC has the potential to be different between the UKB and ABCD cohorts due to age - white matter volume is known to increase through childhood and start decreasing in middle adulthood[41], as well as the smaller sample size in ABCD making it harder for LDSC to detect polygenic effects[42]. To complement these estimates and leverage the availability of individual-level data and greater statistical power, we also employed the bivariate GREML approach in GCTA (*--reml-bivar*), using the AI-REML algorithm while controlling for age, sex and age*sex, and testing for significance of the genetic correlation against the hypothesis that the genetic correlation is 0 (*--reml-bivar-lrt-rg 0*)[40].

## Gene-mapping and gene enrichment analyses

Genetic variants (SNPs) were mapped to genes using information about genomic position, expression quantitative trait loci (eQTL) information, and 3D chromatin interaction mapping as implemented in FUMA v1.5.2 with the experiment-wide significance threshold ($p = 6.13 \times 10^{-9}$)[146]. Pathway enrichment analyses using the results from the full meta-analyses with no pre-selection of genes via MAGMA v1.08[45] gene-set analysis in FUMA. Genes located in the MHC region were excluded (hg19: chromosome 6: 26 Mb–34 Mb). There were 19,021 gene sets from MSigDB v7.0[147] (Curated gene sets: 5500, GO terms: 9996), and 9 other data resources including KEGG, Reactome, and Biocarta (https://www.gsea-msigdb.org/gsea/msigdb/collection_details.jsp#C2). MAGMA uses gene-based P-values to identify genes that are more strongly associated with a phenotype than would be expected by chance. MAGMA then applies a competitive test to compare the association of genes in a gene set to the association of genes outside of the gene set. This allows MAGMA to identify gene sets that are enriched for association signals. MAGMA corrects for a number of confounding factors, such as gene length and size of the gene set, to ensure that the results are not due to chance. A gene-based association analysis (GWGAS) in MAGMA was completed using the full summary statistics for each trait from METAL. Corrections for multiple comparisons were completed using the Bonferroni approach.

To determine whether genes associated with CC morphometry cluster into biological functions, tissue types, or specific cell types, we used the full results of the meta-analyzed genome-wide association studies (GWAS) rather than prioritizing genes. Pathway analysis, as described above, was completed.

We performed gene-property and gene-set analysis using the MAGMA software on 54 tissue types from the GTEx v8 database and BrainSpan[47,48], which includes 29 samples from individuals representing 29 different ages of brains, as well as 11 general developmental stages.

Single cell RNA-sequencing data sets used in the cell-type specific analyses included the human developmental and adult brain samples from the PsychENCODE consortium[148], human brain samples of the middle temporal gyrus and lateral geniculate nucleus from the Allen Brain Atlas[149], human brain samples using DroNc-seq[150], two datasets of human prefrontal cortex brain samples across developmental stages which show per cell type average across different ages, and per cell type per age average expression[151], two datasets of human brain samples with and without fetal tissue[152], human brain samples from the temporal cortex[153], and human samples from the ventral midbrain from 6–11 week old embryos[154]. A 3-step workflow is implemented in FUMA to determine the association between cell-type-specific expression and CC morphometry-gene association supported by multiple independent datasets, which has been extensively described[50]. All tests were corrected using the Bonferroni approach.

## Partitioned heritability of meta-analysis results by cell and tissue type with LDSC

Partitioned heritability analysis was completed to estimate the amount of heritability explained by annotated regions of the genome[46,49]. We tested for enrichment of CC $h^2$ of variants located in multiple tissues and cell types using the LDSC-SEG approach, with all analyses being corrected for the FDR[49]. Annotations indicating specific gene expression in multiple tissues/cell types from the Genotype-Tissue Expression (GTEx) project and Franke lab were downloaded from https://alkesgroup.broadinstitute.org/LDSCORE/LDSC_SEG_ldscores/. We also downloaded 489 tissue-specific chromatin-based annotations from narrow peaks for six epigenetic marks from the Roadmap Epigenomics and ENCODE projects[155,156]. These annotations were downloaded from the URL mentioned above. This would allow us to either verify or identify new findings from the gene expression analysis from an independent source using a different type of data. Finding new patterns of chromatin enrichment can help us to understand how genes are regulated. For example, if we find that a particular epigenetic mark is enriched in a region of the genome that is associated with a specific gene in a specific tissue type, this could suggest that the gene is regulated by that epigenetic mark in that specific tissue type. Gene expression data from the Immunological Genome (ImmGen) project[157], which contains microarray data on 292 immune cell types from mice, was used to test immune cell-type-specific enrichments. Data was downloaded from the aforementioned link.

## LAVA - TWAS

We used the LAVA-TWAS framework to investigate the relationship between CC traits and gene expression in brain tissues, fibroblasts, lymphocytes, and whole blood from the GTEx consortium (v8)[158] in all protein coding genes, as it has ability to model the uncertainty of eQTL effects compared to other commonly used TWAS approaches, which have been shown to be prone to high type-I errors (false positives), and

provides a directly interpretable effect size in the $r_G$ estimate[51]. Analyses were performed on all protein-coding genes (N = 18,380) between all CC phenotypes and eQTLs/sQTLs for each tissue. Genotype data from the European sample of the 1000 Genomes (phase 3) project[159] was used to estimate SNP LD for LAVA. For each eQTL/sQTL that had a significant genetic signal for both the CC phenotype and cortical phenotype (univariate $p$-values less than $1 \times 10^{-4}$), the local bivariate genetic correlation between the two was estimated and tested. All LAVA-TWAS results were corrected using the Bonferroni approach. Following TWAS, trait-specific enrichment analysis via a Fisher's exact test of the top 1% of genes, to evaluate over-representation in 7246 MSigDB v6.2[160] gene sets and gain insight into biological pathways, was conducted. Gene sets were subset such that they must have consisted of at least one of the top 1% of genes, to avoid testing gene-sets with no significantly associated genes. All enrichment testing for eQTLs and sQTLs was performed with Bonferroni correction for every test conducted across all CC phenotypes.

### Global and local genetic correlations with cortical morphometry and mendelian randomization

The CC develops in such a manner that callosal projections are over-produced then refined during development. The majority of cortical projections are refined during postnatal stages and are under the influence of guidance cues[6]. As many genes are responsible for callosal axon guidance, we sought to investigate the genetic relationship between our derived CC traits and the genetic architecture of the human cerebral cortex[6]. We used LDSC to determine the global genetic correlation between area and thickness of the total and parcellated regions of the CC, and the GWAS summary statistics of each globally corrected region-of-interest of the cerebral cortex from the ENIGMA-3 GWAS[161]. We performed bi-directional Mendelian Randomization analyses to investigate if significant genetic correlations observed could be driven by genetic causal relationships between an exposure (e.g., area and thickness of different regions of the CC) and outcome (e.g., regional surface area & cortical thickness). Analyses were performed with summary statistics using GSMR[52]. GSMR includes an integrated HEIDI-outlier feature to detect and remove pleiotropic SNPs. Even if the small-effect pleiotropic SNPs persist, the estimate remains unaffected by pleiotropy[52]. Additionally, GSMR has been extensively shown to be robust to sample overlap between exposure and outcome variables[162]. All analyses were corrected using the Bonferroni approach. To capture potential local shared genetic effects across the genome, we ran LAVA[28] for all protein-coding genes (N = 18,380) between all CC phenotypes and surface area and cortical thickness of regions in the ENIGMA3 GWAS. Genotype data from the European sample of the 1000 Genomes (phase 3) project[159] was used to estimate SNP LD for LAVA. Sample overlap was estimated using the intercepts from bivariate LDSC and integrated into the analysis[28,163]. For each gene that had a significant genetic signal for both the CC phenotype and cortical phenotype (univariate $p$-values less than $1 \times 10^{-4}$), the local bivariate genetic correlation between the two was estimated and tested. All results were corrected using the Bonferroni approach.

### Global and local genetic correlations with neuropsychiatric conditions and mendelian randomization

Abnormalities of the corpus callosum (CC) have been widely implicated in various neurological and neuropsychiatric conditions. To investigate shared genetic architecture, we selected 22 traits for genetic correlation analyses based on their relevance to CC-related pathology and the availability of well-powered GWAS summary statistics, all of which have active ENIGMA working groups[78]. These traits include Alzheimer's disease (AD)[164], attention deficit-hyperactive disorder (ADHD)[86], autism spectrum disorder (ASD)[165], anorexia[166], anxiety[167], bipolar disorder[168], chronic overlapping pain conditions[169], depression[170], epilepsy[171], intelligence[172], insomnia[173], neuroticism[139],

obsessive-compulsive disorder (OCD)[174], Parkinson's disease (PD)[175], post-traumatc stress disorder (PTSD)[176], panic disorder[177], schizophrenia[178], substance abuse[179], suicide attempt[180], and Tourette's syndrome[181]. We used linkage disequilibrium score regression (LDSC) to assess global genetic correlations between these traits and both the area and thickness of the total and parcellated CC. Mendelian randomization and local genetic correlation analyses were then conducted following the same analytic framework used for cortical brain phenotypes.

### Cross study comparisons

There have been two recent GWASes looking at the volume of the corpus callosum and its subregions[16,17]. Building off the notion that area and thickness of brain phenotypes have distinct genetic influences[25], we aimed to compare the genomic loci discovered in the present study with the previous volume GWASes. Although the previous studies used PLINK[182] in the UKB, we used REGENIE[137], which implements a mixed-model approach to account for potential kinship as 59.3% of individuals in the UKB are at least 5th degree relatives[183,184]. In order to determine distinct genetic loci associated with area vs thickness vs volume, and differing enriched biological pathways, significant genomic loci were obtained from Chen et al.[17] and Campbell et al.[16] SNPs indicating significant genomic loci from both studies were entered into the FUMA platform while using the Bonferroni corrected p-value of $5 \times 10^{-8}/11 = 4.55 \times 10^{-9}$ for Chen et al., and the reported $9.6 \times 10^{-9}$ from Campbell et al. The mapped genes from FUMA for each study were then entered into g:Profiler for a multi-gene-list analysis to determine common and distinct gene ontology categories, biological pathways, and transcription factors[144]. All analyses were completed using the g:SCS threshold[145], all gene ontology categories, all biological pathway categories, and the TRANSFAC database.

### Reporting summary

Further information on research design is available in the Nature Portfolio Reporting Summary linked to this article.

## Data availability

This work is a meta-analysis. The full meta-analytic summary statistics are available on the GWAS Catalog (https://www.ebi.ac.uk/gwas/) using the GCP ID GCP001419, and study accession numbers GCST90672009-GCST90672020.

## Code availability

The code and model used to extract the CC and its metrics is available at https://github.com/USC-LoBeS/smacc/.

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

## Acknowledgements

This work was supported by the National Institutes of Health (Grant Nos. R01MH134004 and R01AG059874 [NJ], RF1NS136995 [PMT and NJ] and R01AG087513 [NJ], National Science Foundation Graduate Research Fellowship Program (Grant No. 2020290241 [RRB], R01MH126213, R01NS105746, the Adolescent Brain Cognitive Development (ABCD) Study (https://abcdstudy.org), and UK Biobank (Resource Application No. 11559). SEM was supported by NHMRC grants APP1172917 and APP1158127. Research reported in this publication was supported by the Office Of The Director, National Institutes Of Health of the National Institutes of Health under Award Number S10OD032285. The content is solely the responsibility of the authors and does not necessarily represent the official views of the National Institutes of Health.

## Author contributions

Conceptualization and design: R.R.B., S.P.G., and N.J.; Genetics Methodology: RRB, SPG, AS, CDL, SEM, PMT, NJ; Imaging Methodology: S.P.G., I.B.G., S.J., A.R., E.N., A.H.Z., R.R.B., and N.J.; Data analysis: R.R.B., S.P.G., A.S., S.J., I.B.G., C.D.L., S.E.M., and N.J.; Visualization: R.R.B., S.P.G., A.S., E.H., and N.J.; Drafting of manuscript: RRB, SPG, NJ; All authors contributed to the critical revision of the manuscript. All figures are original, and there are no copyright restrictions or attribution requirements.

## Competing interests

The authors declare no competing interests.
