## [Transparent Peer Review file · Nature Communications]

The Genetic Architecture of the Human Corpus Callosum and its Subregions

Corresponding Author: Dr Ravi Bhatt

Version 0:

Reviewer comments:

Reviewer #1

(Remarks to the Author)

Line 22 and 25 and 29 and line 36 and 39: Use the abbreviation for corpus callosum CC as you defined it already in line 19

Line 58: Replace 3D with Three-dimensional

Line 87 and line 360: Replace corpus callosum with CC

Line 263 and 277: Add reference for Witelson's article

Line 436: Impulsivity and autism were not part of the results and should be included in the results section

Line 459: Replace corpus callosum with CC

Line 470: Results for IQ was not given in the results section

Line 512: Why were the images rotated in increments of 15 degrees? What was the reason the researchers decided to use 15 degrees?

Line 536: Replace corpus callosum with CC

Line 656 and 657 and 681 and 782: Replace corpus callosum with CC

General comments:

Of which population groups does the non-European populations consist?

In the discussion, refrain from repeating results without an explanation for them

The European vs non-European population groups were not compared (results) or discussed in the manuscript

(Remarks on code availability)

The results of the paper are reproducible although many data repository banks were used, which can become confusing, I could install and run the code

Reviewer #2

(Remarks to the Author)

This manuscript by Bhatt, Gadewar et al. describes a genome-wide association study of the corpus callosum. The authors thereby leveraged a novel approach to extract CC regional morphological features and made use of data from two highly distinct cohorts. This study employs a comprehensive set of post-GWAS analyses that provide insights into the molecular mechanisms that drive CC morphology, as well as its relevance for psychiatric disorders. However, the authors do gloss over quite a few details that are important to understand the novelty, validity and robustness of the findings. Further, I think it would be good for the authors to ground these findings more in the existing literature, especially in light of the other, published GWAS on corpus callosum. There are some interesting findings which are different to these other papers, which I think the authors could expand upon. Here more specific comments in no particular order:

- There have been at least two published large-scale GWAS on the CC and its subregions (Campbell et al., Chen et al.), which made use of the same UKB dataset (~80-90% sample overlap), analysed roughly the same CC metrics, and performed substantially overlapping sets of post-GWAS analyses. While these are cited, they are tucked away in a short statement about heritability in the intro and ignored in other relevant places (like the discussion). I think the comparison should be made more explicit, stating why the current study has added value, e.g. due to the new segmentation approach. In

that light, perhaps the overall results should also be compared to these studies (e.g. # discovered loci, genetic correlations). I think it would be good for the authors to mention how their work is different to that which already exists and to outline what value this GWAS represents to the field.

- The authors refer to the relationship between neuropsychiatric traits and the CC as a primary aim however, there is already work on this. In addition, despite this being mentioned as a primary aim, the relationship with psychiatric traits is mentioned in a very succinct paragraph toward the end of the discussion.
- Perhaps rephrase the text in abstract and intro a bit to remove the implication that the segmentation tool is completely novel; there is a publication on this (which is good, as that provides the necessary details on its development and validation).
- Although another publication exists on the segmentation protocol, I think it would be good to mention how this segmentation approach differs from the existing approaches, where GWAS have already been done in UKB (line 69 to 74)- how does this add to our understanding of the CC.
- I think the replication results are presented a bit too loosely ('we observed concordance and similar effect magnitude'). Please expand.
- Somewhat related to this, I actually find the genetic correlations between cohorts rather low (and with big SE). Just stating in the main text that they are significant is therefore not really doing justice to this, a finding that could be seen as of interest; why is it so low? E.g. age or anything related to UK vs USA? There is a little statement about this in the Methods, but I don't think that suffices.
- The discovery of negative genetic correlations between cortical brain regions and the CC is interesting. It would be valuable for the authors to offer a plausible explanation for this phenomenon. Additionally, I wonder if these same relationships persist when examining correlations based on the thickness or surface area of these regions, rather than solely genetic correlations.
- Line 461 to 471: Some of the BD results are discussed and a plausible explanation is given for this phenotype. However, the same is not done for ADHD. I think in general, since the relationship with neuropsychiatric traits is mentioned as a primary aim- this should be expanded upon. The lack of findings with other psychiatric phenotypes also differs to existing literature, which is not mentioned nor explained.
- The applied multiple comparisons correction for each subanalysis is unclear. The Methods section states at the end of most paragraphs that 'tests were corrected using the Bonferroni approach', but it doesn't provide any details on how many tests were corrected for, i.e. which of the comparisons this approach is considering. This is more of a problem than just a bit of lack of clarity, given the many many tests conducted.
- Have the authors considered controlling for brain global measures such as ICV in their GWAS? At minimal, some discussion about the impact of such a decision (which is very common in this field) is necessary.
- Please confirm whether sample overlap was taken into account for analyses that are sensitive to this, e.g. the Mendelian randomization.
- Discussion, paragraph 1- contrary to what is mentioned, there have been recent large-scale genetic studies on humans, using GWAS in the same dataset as this manuscript (referenced in the intro).
- I think that overall flow could be improved in the discussion- for example there are some very short paragraphs consisting of one or two sentences which could be improved (line 281 to 303). However, more generally, I think that the aim that is mentioned in the introduction is slightly lost in the discussion. Since there have already been large-scale GWAS on this brain region, I think it could be interesting for the authors to compare results and to talk about the differences that they found; Comparing their results to existing literature on the topic and the differences noted in their own manuscript (for example, as mentioned above- the differences noted between UKB and ABCD).

(Remarks on code availability)

Reviewer #3

(Remarks to the Author)

(Remarks on code availability)

Version 1:

Reviewer comments:

Reviewer #1

(Remarks to the Author)

Thank you for addressing my reviewer comments extensively and in detail. The results and discussion sections are significantly improved. I have no further comments.

(Remarks on code availability)

Reviewer #2

(Remarks to the Author)

Thank you for your replies. While the manuscript has improved, I find the edits did not resolve most of the raised issues, as outlined below.

1) The statement that this is the first (meta-analysis) GWAS of the CC is rather misleading. The other two GWAS on the CC, given 90% sample overlap, should not be brushed off by one minor sentence in the Discussion. The reply to one of the comments that the publication of the previous GWAS has been made more clear in the Discussion is also simply not true. Further, the writing that this is the first GWAS looking into thickness is incorrect. Campbell et al., also looked at thickness in their GWAS. It is therefore worth mentioning how these findings differ or how this study provides more meaningful information.

2) Related to the first point, since the inclusion of additional metrics is given as the main motivation for an additional CC GWAS, wouldn't it be meaningful to see if there is overlap between the surface area, thickness and volume results, respectively? How does this provide additional mechanistic insight when this is an association study, as are the others? I am still missing an explanation how the differences in the underlying architecture of SA, TH and volume add to our understanding of the relationship with the CC and psychiatric disorders?.

3) Thank you for adding more information on the replication results. However, there appears to be an issue with the presented numbers, and they are certainly not in line with the conclusions. It is stated 'We observed a concordance in direction and similar magnitude effect sizes in the analyses within the data from the non-European participants. Out of the 152 significant loci identified across CC phenotypes in this study, 78 loci demonstrated a consistent direction of effect between different populations'. This means the concordance (78/152) is virtually a coin toss. That is not at all reflecting decent generalization across the cohorts. Please comment on this.

4) The response to the feedback on the cohort comparisons/correlations does not fully address how the differences between the cohorts (UKB vs. ABCD) might play a role in the observed low correlations. While age differences are considered, other cohort-specific factors (e.g., ethnicity, environmental influences, or recruitment methods) are likely to contribute to the variability in genetic correlations. You may want to highlight whether these factors have been explored adequately or if they require further attention.

Further, it's true that smaller sample sizes can be a limitation with LDSC. However, while the smaller sample size of the ABCD cohort could influence the results, this is unlikely to account for the observed low correlations. Additionally, if the LDSC method is underpowered, it is worth exploring alternative methods to estimate genetic correlations.

5) It is still unclear how traits were selected for the genetic correlation analyses. The correlations (and lack thereof for some traits) are still not fully explained.

6) Regarding the feedback of missing info on multiple comparisons correction, the reply is that the Bonferroni corrections are named at each Supplementary Table. This is not sufficient, and it remains still unclear how these corrections were chosen. This is very important. For instance, from the LAVA example included, there appears to be no correction for the number of traits that were tested. This would absolutely be necessary, it cannot be avoided.

7) The response to the feedback on inclusion of an ICV covariate is minimal. What is the overall observed pattern comparing results with and without ICV, and how is this interpreted?

As a final general point, please consider to more precisely indicate what is truly revised text in future rebuttals to reviewers. A large proportion of the green text (stated to reflect modified text) was not novel, with perhaps here and there one word in a sentence added. This is misleading. I also find the replies rather brief

(Remarks on code availability)

Reviewer #3

(Remarks to the Author)

(Remarks on code availability)

Version 2:

Reviewer comments:

Reviewer #2

(Remarks to the Author)

Thank you for the replies. I have no further comments

(Remarks on code availability)

Reviewer #3

(Remarks to the Author)

(Remarks on code availability)

REVIEWER COMMENTS

Reviewer #1 (Remarks to the Author):

Line 22, 25, 29, 87, 360, 36, 39, 459, 536, 656 and 657 and 681 and 782: Use the abbreviation for corpus callosum CC as you defined it already in line 19

We have made sure to use CC as the abbreviation for corpus callosum throughout the manuscript following the first mention of it.

Line 58: Replace 3D with Three-dimensional

Thank you. We have replaced 3D with three-dimensional.

Line 263 and 277: Add reference for Witelson's article

Thank you. We have added the reference for Witelson atlas in places throughout the manuscript where appropriate.

"We conducted a GWAS of area and mean thickness of the whole CC, and five regions of the Witelson parcellation scheme (Fig. 1)^{23,24},"

"When parcellating by the Witelson scheme²³,"

"When analyzing the regional Witelson parcellations²³,"

"The regional shape metrics were based on a 5 compartment version of the Witelson atlas^{23,24}. The Witelson atlas is composed of the (1) genu, (2) anterior midbody, (3) posterior midbody, (4) isthmus, and (5) splenium. The metrics used for the GWAS analysis were area and mean thickness of the total CC and all of the parcellations of the Witelson atlas.

Witelson, S. F. Hand and sex differences in the isthmus and genu of the human corpus callosum. A postmortem morphological study. *Brain* **112 (Pt 3)**, 799–835 (1989).

Line 436: Impulsivity and autism were not part of the results and should be included in the results section

This line read “*clinical phenotypes associated with the central nervous system due to copy number variations of chr10q26.13 include abnormal cranium development, global developmental delay and learning difficulties, and neuropsychiatric manifestations including ADHD, impulsivity or autistic behaviors*” and was in reference to the local genetic correlations observed at the chr10q26.13 cytogenetic band between the splenium and isthmus thickness, and the thickness of the superior parietal cortex. The chr10q26.13 has been associated with many neurological clinical phenotypes as expanded upon in this section, all of which have research implicating the role of the superior parietal cortex. This is not in reference to our tests for the aforementioned phenotypes.

Before: “Clinical phenotypes associated with the central nervous system due to copy number variations of chr10q26.13 include abnormal cranium development, global developmental delay and learning difficulties, and neuropsychiatric manifestations including ADHD, impulsivity or autistic behaviors^{67–69}.”

We have made this more clear in the text:

After: “Previous clinical neuropsychiatric conditions associated with copy number variations of chr10q26.13 include abnormal cranium development, global developmental delay and learning difficulties, and neurodevelopmental manifestations including ADHD or autistic behaviors^{69–71}.”

Line 470: Results for IQ was not given in the results section

After redoing the local genetic correlation analysis after including more traits as suggested by reviewer 2, and incorporating sample size of each GWAS to get more accurate estimates of marginal SNP effects, the results for IQ were no longer significant after controlling for multiple comparisons. Thus, we have removed this from the results and discussion. However, all the results for every trait tested are included in the supplementary tables.

Line 512: Why were the images rotated in increments of 15 degrees? What was the reason the researchers decided to use 15 degrees?

Thank you for the comment. We used data augmentation techniques to increase the number of samples in training by making small changes to the data to include small variations in data and make the model more robust to these variations. We have added the following text in the manuscript:

“The original images in the training set already had 5-10 degrees rotation variation, so we rotated images in increments of 15 degrees to include more variety of head orientations.”

General comments:

Of which population groups does the non-European populations consist?

We have placed tables of the ancestry of all non-European individuals used on the non-European GWAS in the supplementary tables (Table S36). The ancestry composition was determined using the KING software and the Hapmap3 reference panel.

We have also added in the manuscript details about how the ancestry was determined.

*"We also analyzed non-European ancestry individuals to examine the generalization of the observed effects across ancestries. In order to accurately estimate non-European individuals using the HapMap3 reference panel, the KING software package¹¹⁵, which uses a well validated MDS and support vector machine approach, was used to estimate ancestry composition in all individuals not used in the principal GWAS (**Supplementary Table 36**). While making sure no individuals classified as CEU, TSI, or a combination of both, were not included, we included 636 individuals from the UKB and 4129 individuals from ABCD."*

UK Biobank	N = 636 [†]
Ancestry	
ASW	38 (6.0%)
CEU;GIH	28 (4.4%)
CEU;MEX	21 (3.3%)
GIH	404 (64%)
GIH;CEU	18 (2.8%)
MEX	61 (9.6%)

MEX;CEU	13 (2.0%)
MEX;JPT	13 (2.0%)
Other mixed trace ancestries	40 (5.9%)

¹n (%)

ABCD	N = 4,129¹
Ancestry	
ASW	1,249 (30%)
ASW;CEU	195 (4.7%)
ASW;YRI	86 (2.1%)
CEU;ASW	80 (1.9%)
CEU;CHD;TSI	56 (1.4%)
CEU;MEX	118 (2.9%)
CHD	142 (3.4%)
GIH	42 (1.0%)
MEX	1,241 (30%)
MEX;CEU	91 (2.2%)
MEX;TSI	87 (2.1%)

TSI;MEX	86 (2.1%)
YRI	121 (2.9%)
YRI;ASW	40 (1.0%)
Other mixed trace ancestries	495 (12.3%)

¹n (%)

Ancestry composition of all "non-European" individuals in UK Biobank and ABCD used for the CC GWAS as determined by KING. The HapMap3 Ancestry Panel was used. The number of individuals and percentage of the whole population sample is shown. ASW – African ancestry in Southwest USA, CEU – Utah residents with Northern and Western European ancestry from the CEPH collection, CHB – Han Chinese in Beijing, China, CHD – Chinese in Metropolitan Denver, Colorado, GIH – Gujarati Indians in Houston, Texas, JPT – Japanese in Tokyo, Japan, LWK – Luhya in Webuye, Kenya, MXL – Mexican ancestry in Los Angeles, California, MKK – Maasai in Kinyawa, Kenya, TSI – Toscani in Italia, YRI – Yoruba in Ibadan, Nigeria. Individuals who have multiple mixed ancestries, but are in a group making up less than 1% of the sample, are labelled as "Other mixed trace ancestries"

In the discussion, refrain from repeating results without an explanation for them

Thank you for pointing out the need to avoid repetition of results without providing explanations in the Discussion section. We have carefully revised the discussion including the revisions requested by the reviewers, and ensured results are not repeated without interpretation. We have modified the Discussion section (in green):

We conducted the first GWAS meta-analysis of CC morphometry across two cohorts with vastly different age ranges, leveraging our artificial intelligence-based tool, SMACC, to extract detailed CC phenotypes from 46,685 individuals across the UKB and ABCD studies. While prior research into the genetic basis of CC structure and development has primarily relied on candidate gene approaches in animal models and post-mortem human studies, our work addresses the notable

differences between the human CC and its counterparts in animal models⁶. This study offers genome-wide insights into the genetic architecture of the human CC in vivo, significantly advancing our understanding of its development and variation. Previous GWAS efforts focused on CC volume using FreeSurfer-derived measures solely in the UKB cohort^{16,17}. Here, we expand on the well-replicated finding that area and thickness measures of neuroimaging phenotypes have distinct genetic influences²⁵. Our findings strongly support this distinction, as our meta-analysis revealed zero overlapping significant loci between the area and thickness phenotypes of the CC. This underscores the value of separating these metrics to identify unique genetic contributions to CC morphometry. Furthermore, while previous studies have reported genetic correlations between CC volume and neuropsychiatric traits such as bipolar disorder and ADHD^{16,17}, our investigation extends these findings by exploring the specific genetic influences on CC area and thickness. This approach enables a deeper understanding of the mechanistic underpinnings behind these associations. Notably, we identified localized, distinct genetic relationships between CC morphometry and traits like neuroticism and Tourette's syndrome—associations that had not been previously reported.

We show the genetic architecture of the CC is highly polygenic, and specific genetic variants influence CC subregions along a rostral-caudal gradient. Five loci that were positionally mapped to genes were identified to influence both total area and mean thickness of the CC (IQCJ-SHIP1, FIP1L1, HBEGF, CDKN2B-AS1, and FAM107B). IQCJ-SHIP1 had the strongest effect across total area and mean thickness, implicating mechanisms such as conduction of action potentials in myelinated cells via organizing molecular complexes at the nodes of Ranvier and axon initial segments, calcium mediated responses, as well as axon outgrowth and guidance⁵⁰. The strongest locus for total area was mapped to the STRN gene. STRN has been heavily implicated in the Wnt signaling pathway, which controls the expression of genes that are essential for cell proliferation, survival, differentiation, and migration via transcription factors^{51–53}. The HBEGF gene was the strongest locus for total mean thickness, implicating mechanisms in early development. HBEGF expression is localized in the ventricular zone and cortical layers during development⁵⁴, and has been implicated in regulating cell migration via chemoattractive mechanisms⁵⁴. Significant enrichment of heritability of total mean thickness in various histone marks from chromatin data (ATAC-seq) of the fetal brain and cortex derived primary cultured neurospheres, significant tissue expression in the brain 19-weeks post conception, as well as enrichment of gene sets involving regulation of histone modification, suggests genetic variants in regions of open chromatin and

transcriptional activity regulation in early development are key mechanisms underlying CC morphometry. When histones are acetylated, they become more negatively charged. This negative charge repels the negatively charged DNA, causing the DNA to be “pushed away” from the histones. This loosening of the DNA-histone complex makes it easier for transcription factors to access the DNA and initiate transcription⁵⁵.

Parcellation of the CC into the five regions defined by the Witelson scheme allowed for further refinement and genetic understanding of its morphometry in a rostral-caudal gradient. Our results provide insight as to which molecular mechanisms influence this functionally defined gradient (i.e. prefrontal, premotor/supplementary motor, primary motor, primary sensory, and parietal/temporal/occipital)²⁴. An overlap of genetic loci along the most anterior (genu and anterior body, SHTN1) and most posterior (isthmus and splenium, IQCJ-SCHIP1) regions of the CC, along with splenium heritability enrichment of in histone chromatin marks of the fetal brain and dorsolateral prefrontal cortex, implicates regulation of neuron migration and action potential conduction. But the overlap of the FOXO3 along the area of the posterior body and isthmus implicates IL-9 signaling and FOXO-mediated transcription responsible for triggering apoptosis⁵⁶. Only the posterior body and isthmus showed heritability enrichment in immune cells including myeloid cells and innate lymphocytes. The thinning of the CC (along the posterior body and isthmus) occurs in a functional gradient connecting the somatosensory and parietal association areas of the brain^{6,57,58}. This follows activity dependent pruning by functional area⁶, where somatosensory circuits are pruned in early development in an experience dependent context⁵⁹. As immune cells are increasingly being recognized as key players in brain maturation and neurodevelopment⁶⁰, our results suggest IL-9 mediating a neuroprotective effect in the CC during the cell dieback phase^{60,61}, and may play a significant role in posterior CC morphometry. LAVA-TWAS results showed another potential mechanism of isthmus pruning via expression of ATP13A2 in fibroblasts, and splicing of genes involved in NF-κB signaling⁶². ATP13A2 is involved in lysosomal-mediated apoptosis⁶³, suggesting such regulation of fibroblast mediated growth of callosal projections⁶⁴. This is also supported by the current discovery of enrichment of genes related to isthmus area in the “regulation of chaperone mediated autophagy pathway”, which may influence isthmus morphometry.

The topographic organization of the CC correlates with the homotopic bilateral regions of the cortex it is known to connect⁵. A variety of genetically regulated principal mechanisms influence CC neuronal and glial proliferation, neuronal migration and specification, midline patterning, axonal growth and guidance, and post-guidance refinement to homotopic analogs in the cortex^{65,66}. Our results suggest potential genetic mechanisms contributing to callosal-cortical organization. We

show an overall negative global genetic correlation of CC phenotypes with the cortical thickness of the cingulate and surface area of the posterior parietal cortices, including a unidirectional negative effect of genu area on rostral anterior cingulate thickness, and total area on precuneus surface area free of any non-genetic confounders. Positive global genetic correlations of total CC area and splenium thickness with cortical thickness in the occipital cortex were also observed. Local genetic correlations of the CC were observed throughout the cerebral cortex, most pronounced with total CC area and splenium thickness. Notable findings included numerous genes in the chr2p22 cytogenetic band showing negative correlations between total CC and posterior body area with precuneus surface area, including the significant STRN gene observed across all CC phenotypes, further implicating the Wnt signaling pathway and dendritic calcium signaling in the context of neurodevelopment^{67,68}. Within this cytogenetic band, HEATR5B was also positively genetically associated with precentral gyrus surface area. Opposing genetic effects were observed between splenium thickness with isthmus cingulate thickness (i.e. positive) vs. superior parietal cortex thickness (i.e. negative) in genes in the chr10q26.13 cytogenetic band. *Previous clinical neuropsychiatric conditions associated with copy number variations of chr10q26.13 include abnormal cranium development, global developmental delay and learning difficulties, and neurodevelopmental manifestations including ADHD or autistic behaviors⁶⁹⁻⁷¹. This provides a novel testable hypothesis for functional follow up studies, as alterations in the isthmus cingulate and superior parietal cortex have been observed in large-scale studies of various neurodevelopmental disorders⁷². Positive genetic associations in the chr19p13.3 cytogenetic band were observed between the isthmus area and isthmus cingulate cortical thickness, which has been implicated with microcephaly, ventriculomegaly and developmental delay^{73,74}.*

Our results demonstrate opposing genetic relationships between CC phenotypes (*area and thickness of the entire CC and its subregions*) and thickness of the cingulate cortex (negative) vs the neocortex (positive), which suggests a strong genetic component underlying the development of the CC via pioneer axons and chemotaxis. *Coupled with the observed negative phenotypic correlations (Supplementary Table 13), this suggests that the relationship between the CC and the thickness of the cingulate cortex (but not surface area) is influenced by distinct genetic mechanisms that govern their development. Developmentally, pioneer axons emerge in the cingulate and project their axons across the midline using guidance cues. A large portion of these callosal projections are pruned and myelinated in an activity dependent manner, such that axonal remodeling is highly dependent on correlated neural activity in the cortex^{6,75-77}. The strongest local genetic correlation supporting this finding was observed between total mean thickness of the CC and rostral anterior cingulate thickness on TGIF1. As TGIF1 is implicated in holoprosencephaly (i.e. where the brain*

fails to develop two hemispheres), forebrain development via alterations in the Sonic Hedgehog (SHH) pathway, and disruption of axonal guidance via chemoattractive mechanisms^{78,79}, these results provide a potential genetic localization for functional follow-up. The isthmus cingulate, in relation to the isthmus and splenium, was the only cingulate region showing positive local genetic correlations, providing further evidence of distinct molecular mechanisms (e.g. immune-mediated apoptosis and regulation of callosal projections) compared to the rest of the CC underlying its structure and development.

Abnormalities of the CC have also been associated with various neurological/neuropsychiatric disorders⁶. This is the first study to demonstrate a significant negative genetic relationship between the CC and ADHD (utilizing the latest ADHD GWAS findings)⁸⁰, and also replicates previously observed negative genetic associations with bipolar disorder.^{16,17} It is important to note, that prior studies focused on brain volume phenotypes, whereas the current study examines area and thickness, which are known to be influenced by different genetic factors.²⁵ The negative global genetic correlations observed in CC area with ADHD and CC thickness with bipolar disorder, indicate that the allelic differences resulting in smaller CC area and thickness are partly shared with those resulting in a greater risk for ADHD and bipolar disorder, respectively. Further evidence of the negative genetic relationship between ADHD and CC area is provided by studies that show the CC is smaller in individuals with ADHD across various ages^{81–83}, suggesting that impaired inter-hemispheric communication between sensorimotor and attentional systems may contribute to symptoms of hyperactivity, impulsivity, and inattention. Our results also provide a credence to future studies investigating the genetic relationship between the CC and bipolar disorder, as differences in the CC in bipolar disorder have been well established^{13,84,85}. Negative local genetic correlations on the 17q21.31 cytogenetic band between genu area and neuroticism implicated the closely located CRHR1 and KANSL1 genes, which were also highly significant genes observed with genu area and splicing QTLs (sQTLs) in various cortical and subcortical tissue types in the TWAS analysis. Neuroticism, a construct historically describing a cluster of negative emotions, thoughts, and behaviors under the umbrella of “negative affect,” is increasingly recognized as a significant predictor of susceptibility to stress-related psychiatric disorders, including anxiety and depression⁸⁶. CRHR1 encodes the corticotropin-releasing hormone receptor 1, a receptor widely expressed in the cortex and central nervous system that mediates the effects of corticotropin-releasing factor (CRF). The CRF system plays a central role in orchestrating the body’s stress response through the hypothalamic-pituitary-adrenal (HPA) axis and autonomic nervous system. Dysregulation of this system has been extensively linked to the pathophysiology of stress-related anxiety and mood disorders, with neuroticism often serving as a measurable intermediate phenotype⁸⁶. CRHR1

polymorphisms have been associated with differential responses to stress and heightened susceptibility to psychiatric disorders⁸⁶, potentially mediated via altered connectivity between the prefrontal cortices via the genu and altered RNA splicing of the CRHR1 gene inside cortical tissue. A positive local genetic correlation was observed between Tourette's syndrome (TS) and the DFFB gene for the cortical thickness of the anterior body of the corpus callosum, which connects callosal fibers to premotor and supplementary motor cortical areas²⁴. DFFB is involved in apoptosis during development, a process critical for proper neural pruning and brain maturation. Disruptions in this gene may influence the structural and functional integrity of the corpus callosum, which could have downstream effects on motor planning and execution. Individuals with TS have been reported to exhibit larger corpus callosum morphometry compared to neurotypical controls, and greater CC size has also been positively correlated with tic severity, suggesting that alterations in callosal morphology may play a role in the pathophysiology of TS^{87,88}. The anterior body, in particular, is critical for coordinating motor functions and integrating cortical activity across hemispheres, regions closely tied to tic generation and suppression^{24,89,90}.

In summary, this work identifies genome-wide significant loci of morphometry of the overall CC and its sectors, convergence on biological functions with a particular importance of apoptosis and pruning during development, tissues and cell types, as well as the genetic overlap with the cerebral cortex and neuropsychiatric conditions.

The European vs non-European population groups were not compared (results) or discussed in the manuscript

We have expanded on this section to show empirically how the effects are similar/different of significant loci between European and non-European results. In Supplementary Table 1, we have first expanded the table to add 95% confidence intervals of the effect sizes. Based on this, we have shown the number of loci in the European results are in the 95% confidence interval of the non-European cohort, as our group has previously done, to show similar direction and magnitude of effect sizes (Grasby, Jahanshad et al. 2020).

*"We observed a concordance in direction and similar magnitude effect sizes in the analyses within the data from the non-European participants. Out of the 152 significant loci identified across CC phenotypes in this study, 78 loci demonstrated a consistent direction of effect between different populations. Furthermore, for 124 of these loci, the effect sizes observed in European participants fell within the 95% confidence interval of those seen in non-European participants. Detailed annotations and regional association plots of all genomic loci, independent significant SNPs and genes are in **Supplementary Tables S1-S4 and Extended Data 1.**"* We show a screenshot of Supplementary Table 1 with the two columns highlighted below showing if the effect

This manuscript by Bhatt, Gadewar et al. describes a genome-wide association study of the corpus callosum. The authors thereby leveraged a novel approach to extract CC regional morphological features and made use of data from two highly distinct cohorts. This study employs a comprehensive set of post-GWAS analyses that provide insights into the molecular mechanisms that drive CC morphology, as well as its relevance for psychiatric disorders. However, the authors do gloss over quite a few details that are important to understand the novelty, validity and robustness of the findings. Further, I think it would be good for the authors to ground these findings more in the existing literature, especially in light of the other, published GWAS on corpus callosum. There are some interesting findings which are different to these other papers, which I think the authors could expand upon. Here more specific comments in no particular order:

- There have been at least two published large-scale GWAS on the CC and its subregions (Campbell et al., Chen et al.), which made use of the same UKB dataset (~80-90% sample overlap), analysed roughly the same CC metrics, and performed substantially overlapping sets of post-GWAS analyses. While these are cited, they are tucked away in a short statement about heritability in the intro and ignored in other relevant places (like the discussion). I think the comparison should be made more explicit, stating why the current study has added value, e.g. due to the new segmentation approach. In that light, perhaps the overall results should also be compared to these studies (e.g. # discovered loci, genetic correlations). I think it would be good for the authors to mention how their work is different to that which already exists and to outline what value this GWAS represents to the field.

We agree with the reviewer and have added information in the manuscript on how the current GWAS is different from previous ones:

"We conducted the first GWAS meta-analysis of CC morphometry, leveraging our artificial intelligence-based tool, SMACC, to extract detailed CC phenotypes from 46,685 individuals across the UKB and ABCD studies. While prior research into the genetic basis of CC structure and development has primarily relied on candidate gene approaches in animal models and post-mortem human studies, our work addresses the notable differences between the human CC and its counterparts in animal models⁶. This study offers genome-wide insights into the genetic architecture of the human CC in vivo, significantly advancing our understanding of its development and variation. Previous GWAS efforts focused on CC volume using FreeSurfer-derived measures solely in the UKB cohort^{16,17}. Here, we expand on the well-replicated finding that area and thickness measures of neuroimaging phenotypes have distinct genetic influences²⁵. Our findings strongly support this distinction, as our meta-analysis revealed zero overlapping significant loci between the area and thickness phenotypes of the CC. This underscores the value of separating these metrics to identify unique genetic contributions to CC morphometry. Furthermore, while previous studies have reported genetic correlations between CC volume and

neuropsychiatric traits such as bipolar disorder and ADHD^{16,17}, our investigation extends these findings by exploring the specific genetic influences on CC area and thickness. This approach enables a deeper understanding of the mechanistic underpinnings behind these associations. Notably, we identified localized, distinct genetic relationships between CC morphometry and traits like neuroticism and Tourette's syndrome—associations that had not been previously reported."

- The authors refer to the relationship between neuropsychiatric traits and the CC as a primary aim however, there is already work on this. In addition, despite this being mentioned as a primary aim, the relationship with psychiatric traits is mentioned in a very succinct paragraph toward the end of the discussion.

We have redone the analysis with neuropsychiatric traits while including more traits as done in previous CC GWAS studies, as well as more accurately estimating the marginal effect sizes for all SNPs by integrating the sample size of each GWAS study as recommended by the LAVA developers. We have expanded on the neuropsychiatric paragraph in the discussion significantly:

"Abnormalities of the CC have also been associated with various neurological/neuropsychiatric disorders⁶. This was the first study to demonstrate a significant negative genetic relationship between the CC and ADHD (utilizing the latest ADHD GWAS findings)⁸⁰, and also replicates previously observed negative genetic associations with bipolar disorder.^{16,17} It is important to note, however, that prior studies focused on brain volume phenotypes, whereas the current study examines area and thickness, which are known to be influenced by different genetic factors.²⁵ The negative global genetic correlations observed in CC area with ADHD and CC thickness with bipolar disorder, indicate that the allelic differences resulting in smaller CC area and thickness are partly shared with those resulting in a greater risk for ADHD and bipolar disorder, respectively. Further evidence of the negative genetic relationship between ADHD and CC area is provided by studies that show the CC is smaller in individuals with ADHD across various ages⁸¹⁻⁸³, suggesting that impaired inter-hemispheric communication between sensorimotor and attentional systems may contribute to symptoms of hyperactivity, impulsivity, and inattention. Our results also provide a credence to future studies investigating the genetic relationship between the CC and bipolar disorder, as differences in the CC in bipolar disorder have been well established^{13,84,85}. Negative local genetic correlations on the 17q21.31 cytogenetic band between genu area and neuroticism implicated the closely located CRHR1 and KANSL1 genes, which were also highly significant genes observed with genu area and splicing QTLs (sQTLs) in various cortical and subcortical tissue types in the TWAS analysis. Neuroticism, a construct historically describing a cluster of negative emotions, thoughts, and behaviors under the umbrella of "negative affect," is increasingly recognized as a significant predictor of susceptibility to stress-related psychiatric disorders, including anxiety and depression⁸⁶. CRHR1 encodes the corticotropin-releasing hormone receptor 1, a receptor widely expressed in the cortex and central nervous system that mediates the effects of corticotropin-releasing factor (CRF). The CRF system plays a central role in orchestrating the body's stress response through the hypothalamic-pituitary-adrenal (HPA) axis and

autonomic nervous system. Dysregulation of this system has been extensively linked to the pathophysiology of stress-related anxiety and mood disorders, with neuroticism often serving as a measurable intermediate phenotype⁸⁶. CRHR1 polymorphisms have been associated with differential responses to stress and heightened susceptibility to psychiatric disorders⁸⁶, potentially mediated via altered connectivity between the prefrontal cortices via the genu and altered RNA splicing of the CRHR1 gene inside cortical tissue. A positive local genetic correlation was observed between Tourette's syndrome (TS) and the DFFB gene for the cortical thickness of the anterior body of the corpus callosum, which connects callosal fibers to premotor and supplementary motor cortical areas²⁴. DFFB is involved in apoptosis during development, a process critical for proper neural pruning and brain maturation. Disruptions in this gene may influence the structural and functional integrity of the corpus callosum, which could have downstream effects on motor planning and execution. Individuals with TS have been reported to exhibit larger corpus callosum morphometry compared to neurotypical controls, and greater CC size has also been positively correlated with tic severity, suggesting that alterations in callosal morphology may play a role in the pathophysiology of TS^{87,88}. The anterior body, in particular, is critical for coordinating motor functions and integrating cortical activity across hemispheres, regions closely tied to tic generation and suppression^{24,89,90}."

Figure 6: The genetic overlap of the corpus callosum and neuropsychiatric phenotypes. (A) Global genetic correlations between CC traits and neuropsychiatric phenotypes. Significant results are designated by the * at the Bonferroni significance threshold of $p = 0.0019$. Significant negative genetic correlations are observed between total and splenium thickness, and bipolar disorder (I). Significant negative genetic correlations are also observed with CC area phenotypes and ADHD.

Of the significant global genetic correlations, significant Mendelian randomization (GSMR) results are displayed, representing the effect of CC phenotypes on neuropsychiatric phenotypes free of non-genetic confounders. (B) Volcano plots showing degree ($-\log_{10}$ p-values) and direction (rG) of local genetic correlations (LAVA) between neuropsychiatric and CC phenotypes. Significant local negative genetic correlations on the CRHR1, KANSL1 and STH genes are observed between genu area and neuroticism. A significant local positive genetic correlation on the DFFB gene is observed between anterior body thickness and Tourette's syndrome. Phenotypes with significant associations are colored (IQ and bipolar II disorder). Significant genes (Bonferroni significance threshold was set at $p = 2.23 \times 10^{-6}$) across all neuropsychiatric phenotypes are shown. AD: alzheimer's disease, ADHD: attention deficit hyperactivity disorder, ASD: autism spectrum disorder, BD: bipolar disorder, BD-I: bipolar I disorder, BD-II: bipolar II disorder, COPC: chronic overlapping pain conditions, IQ: intelligence quotient, OCD: obsessive-compulsive disorder, PTSD: post-traumatic stress disorder, SCZ: schizophrenia.

- Perhaps rephrase the text in abstract and intro a bit to remove the implication that the segmentation tool is completely novel; there is a publication on this (which is good, as that provides the necessary details on its development and validation).

Before: "To characterize the morphometry of the midsagittal corpus callosum, we developed a publicly available artificial intelligence based tool to extract, parcellate, and calculate its total and regional area and thickness."

We have changed the text in the abstract to say:

*After: "To characterize the morphometry of the midsagittal CC, we used **our** publicly available artificial intelligence based tool to extract, parcellate, and calculate its total and regional area and thickness."*

- Although another publication exists on the segmentation protocol, I think it would be good to mention how this segmentation approach differs from the existing approaches, where GWAS have already been done in UKB (line 69 to 74)- how does this add to our understanding of the CC.

Thank you so much for your comment. We have added the text in the Methods section to explain how our tool is different from the existing ones:

"We developed a UNet based automated segmentation tool that segments mid CC in multiple modalities like T1w, T2 and FLAIR, assesses the quality of the segmentation using machine learning methods on the meaningful metrics extracted from the segmentation and is generalizable

to data from various scanners and sites. To our knowledge, there has been no published integrated pipelines for mid CC extraction with quality control in multiple MR modalities. Existing deep learning method like DeepnCCA has been trained to segment mid CC but only works on T2w images and has been trained on data from one scanner only, so it might not be generalizable to data from other scanners. Other existing methods like FreeSurfer (which was used in previous CC GWAS studies)^{16,17}, FastSurfer and a few UNet based methods⁹¹ segment CC but do not assess the quality of the segmentations.”

- I think the replication results are presented a bit too loosely (‘we observed concordance and similar effect magnitude’). Please expand.

We have expanded on this section to show empirically how the effects of significant loci are similar/different between European and non-European results. In Supplementary Table 1, we have first expanded the table to add 95% confidence intervals of the effect sizes. Based on this, we have shown the number of loci in the European results are in the 95% confidence interval of the non-European cohort, as our group has previously done, to show similar direction and magnitude of effect sizes (Grasby, Jahanshad et al. 2020). We have also shown how many loci are concordant in effect in general. We show a screenshot of Supplementary Table 1 with the two columns highlighted below showing if the effect size of the top SNP in each genomic locus observed in Europeans is within the 95% confidence interval of Non-Europeans, and if the direction of effect is concordant in European and Non-European results. Some columns are hidden in the screenshot below to make sure the results are visible. The full results are in the Supplementary Tables.

*“We observed a concordance in direction and similar magnitude effect sizes in the analyses within the data from the non-European participants. Out of the 152 significant loci identified across CC phenotypes in this study, 78 loci demonstrated a consistent direction of effect between different populations. Furthermore, for 124 of these loci, the effect sizes observed in European participants fell within the 95% confidence interval of those seen in non-European participants. Detailed annotations and regional association plots of all genomic loci, independent significant SNPs and genes are in **Supplementary Tables S1-S4 and Extended Data 1.**”*

Grasby, K. L., Jahanshad, N., Painter, J. N., Colodro-Conde, L., Bralten, J., Hibar, D. P., ... & Van Rooij, D. (2020). The genetic architecture of the human cerebral cortex. *Science*, 367(6484), eaay6690.

signals³⁹. However, strong cross-cohort correlations for total area and isthmus thickness phenotypes suggest that genetic variants affecting these traits are likely consistent across developmental stages."

- The discovery of negative genetic correlations between cortical brain regions and the CC is interesting. It would be valuable for the authors to offer a plausible explanation for this phenomenon. Additionally, I wonder if these same relationships persist when examining correlations based on the thickness or surface area of these regions, rather than solely genetic correlations.

We have now added a paragraph in the discussion which addresses this point. We did find that all phenotypic correlations (Supplementary Table 13) between CC and cortical thickness of the cingulate regions (the posterior cingulate and rostral anterior cingulate) showed negative phenotypic correlations with CC metrics as well (Supplementary Table 13). We also make clear that we ran genetic correlations and phenotypic correlations on cortical surface area and thickness of every ROI with our CC phenotypes (area and thickness of the entire CC and its subregions - genu, anterior body, posterior body, isthmus and splenium).

*"Our results demonstrate opposing genetic relationships between CC phenotypes (area and thickness of the entire CC and its subregions) and thickness of the cingulate cortex (negative) vs the neocortex (positive), which suggests a strong genetic component underlying the development of the CC via pioneer axons and chemotaxis. Coupled with the observed negative phenotypic correlations (Supplementary Table 13), this suggests that the relationship between the CC and the thickness of the cingulate cortex (but not surface area) is influenced by distinct genetic mechanisms that govern their development. Developmentally, pioneer axons emerge in the cingulate and project their axons across the midline using guidance cues. A large portion of these callosal projections are pruned and myelinated in an activity dependent manner, such that axonal remodeling is highly dependent on correlated neural activity in the cortex^{6,75-77}. The strongest local genetic correlation supporting this finding was observed between total mean thickness of the CC and rostral anterior cingulate thickness on *TGIF1*. As *TGIF1* is implicated in holoprosencephaly (i.e. where the brain fails to develop two hemispheres), forebrain development via alterations in the Sonic Hedgehog (*SHH*) pathway, and disruption of axonal guidance via chemoattractive mechanisms^{78,79}, these results provide a potential genetic localization for functional follow-up. The isthmus cingulate, in relation to the isthmus and splenium, was the only cingulate region showing positive local genetic correlations, providing further evidence of distinct molecular mechanisms (e.g. immune-mediated apoptosis and regulation of callosal projections) compared to the rest of the CC underlying its structure and development."*

- Line 461 to 471: Some of the BD results are discussed and a plausible explanation is given for this phenotype. However, the same is not done for ADHD. I think in general, since the relationship with neuropsychiatric traits is mentioned as a primary aim- this should be expanded upon. The lack of findings with other psychiatric phenotypes also differs to existing literature, which is not mentioned nor explained.

We expanded on our findings about the CC area and ADHD, and the similarities/differences of neuropsychiatric traits in the discussion. As done with previous CC GWASes, we have added depression as a trait in our analysis from the newest depression GWAS results from Als et al (2023). We have also added chronic pain, suicide attempt and Tourette's syndrome as traits as these traits have been shown to be associated with CC morphometry in the literature.

Als, T.D., Kurki, M.I., Grove, J. *et al.* Depression pathophysiology, risk prediction of recurrence and comorbid psychiatric disorders using genome-wide analyses. *Nat Med* 29, 1832–1844 (2023). <https://doi.org/10.1038/s41591-023-02352-1>

Samar Khoury, Marc Parisien, Scott J Thompson, Etienne Vachon-Preseu, Mathieu Roy, Amy E Martinsen, Bendik S Winsvold, HUNT All-In Pain, Ingunn P Mundal, John-Anker Zwart, Artur Kania, Jeffrey S Mogil, Luda Diatchenko, Genome-wide analysis identifies impaired axonogenesis in chronic overlapping pain conditions, *Brain*, Volume 145, Issue 3, March 2022, Pages 1111–1123, <https://doi.org/10.1093/brain/awab359>

Mullins, Niamh, et al. "Dissecting the shared genetic architecture of suicide attempt, psychiatric disorders, and known risk factors." *Biological psychiatry* 91.3 (2022): 313-327.

Yu, Dongmei, et al. "Interrogating the genetic determinants of Tourette's syndrome and other tic disorders through genome-wide association studies." *American Journal of Psychiatry* 176.3 (2019): 217-227.

"Abnormalities of the CC have also been associated with various neurological/neuropsychiatric disorders⁶. This is the first study to demonstrate a significant negative genetic relationship between the CC and ADHD (utilizing the latest ADHD GWAS findings)⁸⁰, and also replicates previously observed negative genetic associations with bipolar disorder.^{16,17} It is important to note that prior studies focused on brain volume phenotypes, whereas the current study examines area and thickness, which are known to be influenced by different genetic factors.²⁵ The negative global genetic correlations observed in CC area with ADHD and CC thickness with bipolar disorder, indicate that the allelic differences resulting in smaller CC area and thickness are partly shared with

those resulting in a greater risk for ADHD and bipolar disorder, respectively. Further evidence of the negative genetic relationship between ADHD and CC area is provided by studies that show the CC is smaller in individuals with ADHD across various ages⁸¹⁻⁸³, suggesting that impaired inter-hemispheric communication between sensorimotor and attentional systems may contribute to symptoms of hyperactivity, impulsivity, and inattention. Our results also provide a credence to future studies investigating the genetic relationship between the CC and bipolar disorder, as differences in the CC in bipolar disorder have been well established^{13,84,85}. Negative local genetic correlations on the 17q21.31 cytogenetic band between genu area and neuroticism implicated the closely located CRHR1 and KANSL1 genes, which were also highly significant genes observed with genu area and splicing QTLs (sQTLs) in various cortical and subcortical tissue types in the TWAS analysis. Neuroticism, a construct historically describing a cluster of negative emotions, thoughts, and behaviors under the umbrella of “negative affect,” is increasingly recognized as a significant predictor of susceptibility to stress-related psychiatric disorders, including anxiety and depression⁸⁶. CRHR1 encodes the corticotropin-releasing hormone receptor 1, a receptor widely expressed in the cortex and central nervous system that mediates the effects of corticotropin-releasing factor (CRF). The CRF system plays a central role in orchestrating the body’s stress response through the hypothalamic-pituitary-adrenal (HPA) axis and autonomic nervous system. Dysregulation of this system has been extensively linked to the pathophysiology of stress-related anxiety and mood disorders, with neuroticism often serving as a measurable intermediate phenotype⁸⁶. CRHR1 polymorphisms have been associated with differential responses to stress and heightened susceptibility to psychiatric disorders⁸⁶, potentially mediated via altered connectivity between the prefrontal cortices via the genu and altered RNA splicing of the CRHR1 gene inside cortical tissue. A positive local genetic correlation was observed between Tourette’s syndrome (TS) and the DFFB gene for the cortical thickness of the anterior body of the corpus callosum, which connects callosal fibers to premotor and supplementary motor cortical areas²⁴. DFFB is involved in apoptosis during development, a process critical for proper neural pruning and brain maturation. Disruptions in this gene may influence the structural and functional integrity of the corpus callosum, which could have downstream effects on motor planning and execution. Individuals with TS have been reported to exhibit larger corpus callosum morphometry compared to neurotypical controls, and greater CC size has also been positively correlated with tic severity, suggesting that alterations in callosal morphology may play a role in the pathophysiology of TS^{87,88}. The anterior body, in particular, is critical for coordinating motor functions and integrating cortical activity across hemispheres, regions closely tied to tic generation and suppression^{24,89,90}.”

Figure 6: The genetic overlap of the corpus callosum and neuropsychiatric phenotypes. (A) Global genetic correlations between CC traits and neuropsychiatric phenotypes. Significant results are designated by the * at the Bonferroni significance threshold of $p = 0.0019$. Significant negative genetic correlations are observed between total and splenium thickness, and bipolar disorder (I).

Significant negative genetic correlations are also observed with CC area phenotypes and ADHD. Of the significant global genetic correlations, significant Mendelian randomization (GSMR) results are displayed, representing the effect of CC phenotypes on neuropsychiatric phenotypes free of non-genetic confounders. (B) Volcano plots showing degree ($-\log_{10}$ p-values) and direction (rG) of local genetic correlations (LAVA) between neuropsychiatric and CC phenotypes. Significant local negative genetic correlations on the CRHR1, KANSL1 and STH genes are observed between genu area and neuroticism. A significant local positive genetic correlation on the DFFB gene is observed between anterior body thickness and Tourette's syndrome. Phenotypes with significant associations are colored (IQ and bipolar II disorder). Significant genes (Bonferroni significance threshold was set at $p = 2.23 \times 10^{-6}$) across all neuropsychiatric phenotypes are shown. AD: alzheimer's disease, ADHD: attention deficit hyperactivity disorder, ASD: autism spectrum disorder, BD: bipolar disorder, BD-I: bipolar I disorder, BD-II: bipolar II disorder, COPC: chronic overlapping pain conditions, IQ: intelligence quotient, OCD: obsessive-compulsive disorder, PTSD: post-traumatic stress disorder, SCZ: schizophrenia.

- The applied multiple comparisons correction for each subanalysis is unclear. The Methods section states at the end of most paragraphs that 'tests were corrected using the Bonferroni approach', but it doesn't provide any details on how many tests were corrected for, i.e. which of the comparisons this approach is considering. This is more of a problem than just a bit of lack of clarity, given the many many tests conducted.

We understand that the methods of the paper does not always say the exact number of tests accounted for as we thought it may be repetitive to put in the main manuscript. We have all of the exact values for Bonferroni correction for each analysis in the supplementary tables. For example, to show the corrected value for LAVA results with cortical traits, we have said in Supplementary Table 16:

"significance threshold was set at the Bonferroni corrected level of $0.05/22973 = 2.18 \times 10^{-6}$ "

We have added this in every supplementary table where this is applicable.

- Have the authors considered controlling for brain global measures such as ICV in their GWAS? At minimal, some discussion about the impact of such a decision (which is very common in this field) is necessary.

We have conducted additional analyses where the GWAS was run again with ICV as a covariate. We have provided the results in Supplementary Table 37. We also show the differences between the GWAS results without and with ICV as a covariate in Supplementary Table 38. In this table we show the number of common genomic loci, along with gene names identified of these loci using the FUMA platform between the two analyses.

“To determine if the global measure of brain intracranial volume (ICV) would have an impact on the analysis, all GWAS were completed again using ICV as an additional covariate. All results comparing results without and with ICV as a covariate is shown in Supplementary Table 38.

Table S38: Comparison of Significant Loci in Analyses without and with ICV as a covariate: The number of common and unique genes in each GWAS analyses without and with ICV as a covariate. Genes are identified as the nearest gene of the genomic locus defined using the FUMA platform. ICV = intracranial volume

Trait Name	Number of Common Genes	Number of Unique Genes in noICV	Number of Unique Genes in ICV	Common Genes	Unique to noICV	Unique to ICV
Total Area	14	14	8	STRN, FAM171B, RP11-190P13.2, IQCJ- SCHIP1:IQCJ, FIP1L1, HBEGF, BICD2, FAM107B, NAV2, PPP2R5E, KPNA2, TTC39C:RP11-799B12.2, GAL3ST1, PICK1:RP5-1039K5.13	SDHB, SDCCAG8, AC016727.1, TNIK, CTD-2316B1.1, CTB-118N6.2, FOXO3, SNORA73, RP1-16A9.1, SLC45A4, PLEC, CDKN2B-AS1, KIAA1598, BRSK2	ATP13A2, SDCCAG8:A, KT3, XPO1, SEMA6A, PARP10, ENO4:KIAA1598, MOB2, PCSK6

Total Thickne ss	9	2	0	Y_RNA, STRN, IQCJ- SCHIP1:IQCJ, FIP1L1, HBEGF, PARP10, FAM107B, C16orf95, CABYR	ITGAV:AC017 101.10, CDKN2B-AS1	
Genu Area	11	8	2	STRN, AC016727.1, RPL37A, IQCJ- SCHIP1:IQCJ, AC003084.2, KIAA1598, NAV2, CELF1, NAV3, GAL3ST1, PICK1:RP5- 1039K5.13	ZZZ3, STX6, FAM171B, CTC- 448D22.1, SNORA73, PLEC, CDKN2B-AS1, KANSL1	PARP10, TTC39C:RP1 1-799B12.2
Genu Thickne ss	2	1	1	AC003084.2, CHMP7	MAST4	CELF1
AB Area	9	5	6	STRN, FIP1L1, RNU6-727P, FOXO3, SNORA73, FAM107B, KIAA1598, RP1- 34H18.1, CABYR	SDHB, SOAT1, CPED1, PLEC, DUSP8	GMNC, CLVS1, PARP10, KRTAP5-3, NUP160, CTD- 2302E22.1
AB Thickne ss	4	2	1	STRN, FIP1L1, SNORA73, CABYR	HBEGF, PLEC	PARP10
PB Area	5	3	2	STRN, RP11- 88I21.2, FIP1L1, HBEGF, RP1- 34H18.1	IQCJ- SCHIP1:IQCJ, FOXO3, TTC39C	PARP10, CABYR:RP11 -799B12.4

PB Thickne ss	2	2	0	FIP1L1, TTC39C	PARP10, C16orf95	
Isthmus Area	8	9	1	RP1-37C10.3, STRN, CCDC75P1, TNIK, CTB- 118N6.2, HBEGF, FAM107B:RP11- 7C6.1, C16orf95	IQCJ- SCHIP1:IQCJ, ADD1, TBC1D14, FIP1L1:RP11- 89B16.1, FOXO3, PLEC, KIAA1598, NAV2, CABYR	FIP1L1
Isthmus Thickne ss	7	2	2	RP1-37C10.3, STRN, CCDC75P1, FIP1L1, CTB- 118N6.2, HBEGF, C16orf95	WDPCP, SLC45A4	TNIK, FAM107B:RP 11-7C6.1
Splenu m Area	16	8	7	RP1- 37C10.3:ATP13A 2, AKT3, CCDC75P1, IQCJ- SCHIP1:IQCJ, IQCJ-SCHIP1, TNIK, TBC1D14, FIP1L1, FBXL7, CTB-118N6.2, SEMA3A, BICD2, FAM107B, ENO4:KIAA1598, RP11- 12J10.3:FAM53B, TAOK1	AC007382.1, XPO1, RP11- 493K19.3, STPG2:RP11- 681L8.1:STPG 2-AS1, HBEGF, HGF, BRSK2, RRAS2	CDK15, SEMA3F, RP11- 91A15.1, F11- AS1:RP11- 215A19.2, ANKHD1:AN KHD1- EIF4EBP3:S RA1, PPP2R5E, GAL3ST1

Splenium Thickness	5	4	3	STRN, IQCJ-SCHIP1:IQCJ, TBC1D14, FIP1L1, CTB-118N6.2	ANKRD19P, FAM107B, RP11-12J10.3:FAM53B, RRAS2	CCDC75P1, FAM107B:RP11-7C6.1, METTL10
---	---	---	--	---	---------------------------------------

- Please confirm whether sample overlap was taken into account for analyses that are sensitive to this, e.g. the Mendelian randomization.

The analyses which would potentially be sensitive to sample overlap, GSMR and LAVA, are known to take sample overlap into account and we have expanded on this in the methods:

"GSMR includes an integrated HEIDI-outlier feature to detect and remove pleiotropic SNPs. Even if small-effect pleiotropic SNPs persist, the estimate remains unaffected by pleiotropy⁴⁷. Additionally, GSMR has extensively been shown to be robust to sample overlap between exposure and outcome variables¹³⁷.

To capture potential local shared genetic effects across the genome, we ran LAVA²⁸ for all protein coding genes (N = 18,380) between all CC phenotypes and surface area and cortical thickness of regions in the ENIGMA₃ GWAS. Genotype data from the European sample of the 1000 Genomes (phase 3) project¹³⁴ was used to estimate SNP LD for LAVA. Sample overlap was estimated using the intercepts from bivariate LDSC and integrated into the analysis^{28,138}."

- Discussion, paragraph 1- contrary to what is mentioned, there have been recent large-scale genetic studies on humans, using GWAS in the same dataset as this manuscript (referenced in the intro).

We have made this more clear in the first paragraph of the discussion by stating that it is the first meta-analysis of CC morphometry across vastly different age ranges, and now discuss key phenotypic characteristics which make the current study distinct from past ones.

Before: *“We performed a GWAS meta-analysis of corpus callosum morphometry using our artificial intelligence based extraction tool, SMACC, from 46,685 individuals using UKB and ABCD. The majority of studies investigating the genetic influence via candidate genes on CC structure and development have been conducted using various animal models and post-mortem human studies⁶. Given the difference of the human CC compared to animal models⁶, this study provides genome-wide insight into human variation and genes that influence the human CC in vivo.”*

After: *“We conducted the first GWAS meta-analysis of CC morphometry across two cohorts with vastly different age ranges, leveraging our artificial intelligence-based tool, SMACC, to extract detailed CC phenotypes from 46,685 individuals across the UKB and ABCD studies. While prior research into the genetic basis of CC structure and development has primarily relied on candidate gene approaches in animal models and post-mortem human studies, our work addresses the notable differences between the human CC and its counterparts in animal models⁶. This study offers genome-wide insights into the genetic architecture of the human CC in vivo, significantly advancing our understanding of its development and variation. Previous GWAS efforts focused on CC volume using FreeSurfer-derived measures in the UKB cohort^{16,27}. However, it is well-established that area and thickness measures of neuroimaging phenotypes have distinct genetic influences²⁵. Our findings strongly support this distinction, as our meta-analysis revealed zero overlapping significant loci between the area and thickness phenotypes of the CC. This underscores the value of separating these metrics to identify unique genetic contributions to CC morphometry. Furthermore, while previous studies have reported genetic correlations between CC volume and neuropsychiatric traits such as bipolar disorder and ADHD^{16,27}, our investigation extends these findings by exploring the specific genetic influences on CC area and thickness. This approach enables a deeper understanding of the mechanistic underpinnings behind these associations. Notably, we identified localized, distinct genetic relationships between CC morphometry and traits like neuroticism and Tourette’s syndrome—associations that had not been previously reported.”*

- I think that overall flow could be improved in the discussion- for example there are some very short paragraphs consisting of one or two sentences which could be improved (line 281 to 303). However, more generally, I think that the aim that is mentioned in the introduction is slightly lost in the discussion. Since there have already been large-scale GWAS on this brain region, I think it could be interesting for the authors to compare results and to talk about the differences that they found; Comparing their results to existing literature on the topic and the differences noted in their own manuscript (for example, as mentioned above- the differences noted between UKB and ABCD).

We have greatly expanded on the last paragraph in the discussion based on the reanalysis done with neuropsychiatric traits, which was not greatly expanded on in the previous version of the manuscript. We

have also compared our approach and results with previous GWAS studies as well. We have integrated all the reviewer comments above to make sure there is good flow and paragraph structure, to address comparisons with previous GWAS studies, and expanded on the section about the findings with neuropsychiatric traits, which reinforces one of the main aims of the study. Our discussion in its entirety is included in response to Reviewer 1 Question 8.

Reviewer #3 (Remarks to the Author):

Thank you so much.

Dear Reviewers,

We are grateful for the opportunity to revise our manuscript entitled: "The Genetic Architecture of the Human Corpus Callosum and its Subregions"

for a second round for possible publication in *Nature Communications*.

We thank the reviewers for their thoughtful suggestions and ideas. We have addressed all of the reviewers' comments and questions, which encouraged us to improve the analysis, presentation of our findings, and the discussion. We believe the manuscript has greatly benefitted from the feedback.

Point-by-point revisions in response to individual reviewer comments are below. Changes to the manuscript are shown below in green. The original manuscript is submitted as well. We hope the revised manuscript is now suitable for publication and sincerely appreciate your time and consideration throughout the review process.

1) The statement that this is the first (meta-analysis) GWAS of the CC is rather misleading. The other two GWAS on the CC, given 90% sample overlap, should not be brushed off by one minor sentence in the Discussion. The reply to one of the comments that the publication of the previous GWAS has been made more clear in the Discussion is also simply not true. Further, the writing that this is the first GWAS looking into thickness is incorrect. Campbell et al., also looked at thickness in their GWAS. It is therefore worth mentioning how these findings differ or how this study provides more meaningful information.

Upon contacting the lead author of the Campbell et al publication, the authors verified that univariate statistics of thickness were not included in the published paper. "SM Figure 3 venn" shows the multivariate GWAS results using MOSTest for all 5 subregions using the Hofer segmentation approach for thickness, but the data for which SNPs they were, their alleles, and effect sizes were also not included. They also show two venn diagrams of overlapping loci and genes from the FreeSurfer vs Hofer results. Dr. Campbell confirmed and clarified that they used the software C8 to extract this,

<https://www.nitrc.org/projects/c8c8/>

"but the software was quite buggy and we lost many participants in the QC stage so, as a result, we put our thickness results in the supplementary and focused on the FreeSurfer segmentation."

The authors also provided us with the lead SNPs of the multivariate analysis using thickness from C8, which were not included in the published paper, but these were not directly comparable to our univariate results, and thus we did not proceed further.

The lack of software able to accurately and consistently extract CC traits was a large reason for us to develop our AI based tool, *SMACC*, which we describe in the current manuscript. We show

how it completes more accurate segmentation compared to the widely used FreeSurfer, and does not suffer significant QC issues, as it includes a built-in QC component. According to the additional information provided by the authors, these issues caused a loss of approximately 10,000 participants in the Campbell paper using thickness as a phenotype, which was described in the supplementary information of the paper. We did not face this issue.

We have expanded on our results and discussion by doing a direct comparison between the current study and the studies by Chen et al (2023). and Campbell et al (2023). We have compared the genetic loci from our area and thickness results to the volume results from the published and publicly available GWASes.

There are key methodological differences between the GWAS of the other papers and ours. First, both papers used PLINK to conduct the GWAS in UK Biobank. We opted to use REGENIE, which accounts for kinship and population stratification, and is considered the state-of-the-art approach when conducting a GWAS in large population samples (Mbatchou et al 2021, Nat Gen). It is crucial to use a method accounting for potential kinship as 59.3% of individuals in the UK Biobank are at least 5th degree relatives (Zhang et al. 2025, Cell Rep Methods). We explain this approach in the methods section of the manuscript with appropriate references.

Moreover, Chen et al (2023), used 11 traits in their experiment, but their p-value threshold was not adjusted for multiple traits and stated to be at 5×10^{-8} , which was shown in the tables in the Supplementary Information. Thus, when comparing significant genomic loci hits from our results to theirs, we filtered at the Bonferroni corrected value of $5 \times 10^{-8}/11 = 4.55 \times 10^{-9}$. Similarly, Campbell et al (2023) reported the corrected p-value for their study to be 9.6×10^{-9} . Yet the loci reported in their supplementary tables were GWAS significant for a single trait ($p < 5 \times 10^{-8}$), so we filtered the reported loci to the threshold of $p < 9.6 \times 10^{-9}$ for follow up comparisons.

Both of these studies reported that they used the single trait $p < 5 \times 10^{-8}$ threshold in FUMA, which would not account for the multiple trait comparisons for downstream analysis. We used our experiment-wide threshold of 6.13×10^{-9} in FUMA, to account for any potential false positives. We also used the multiple trait adjusted Bonferroni threshold for each of the volume studies in FUMA when mapping genes to their results.

We mapped the filtered set of multi-trait adjusted significant loci reported for Chen et al and Campbell et al onto genes using FUMA, and compared them to our results for overlap. To understand what potential pathways are different between all four results (CC Area, CC Mean Thickness, CC Volume Chen, CC Volume Campbell) for all of our tested traits, we used the g:Profiler platform to conduct a multi-gene-list analysis to understand pathways specific, and shared by all traits.

We have added information about this comparison to the methods and discussion:

Methods:

“There have been two recent GWASes looking at the volume of the corpus callosum and its subregions^{16,17}. Building off the notion that area and thickness of brain phenotypes have distinct genetic influences²⁵, we aimed to compare the genomic loci discovered in the present study with the previous volume GWASes. Although the previous studies used PLINK¹⁸² in the UKB, we used REGENIE¹³⁷, which implements a mixed-model approach to account for potential kinship as 59.3% of individuals in the UKB are at least 5th degree relatives¹⁸³. In order to determine distinct genetic loci associated with area vs thickness vs volume, and differing enriched biological pathways, significant genomic loci were obtained from Chen et al¹⁷ and Campbell et al¹⁶. SNPs indicating significant genomic loci from both studies were entered into the FUMA platform while using the Bonferroni corrected p-value of $5 \times 10^{-8}/11 = 4.55 \times 10^{-9}$ for Chen et al., and the reported 9.6×10^{-9} from Campbell et al. The mapped genes from FUMA for each study were then entered into g:Profiler for a multi-gene-list analysis to determine common and distinct gene ontology categories, biological pathways and transcription factors¹⁴⁴. All analyses were completed using the g:SCS threshold¹⁴⁵, all gene ontology categories, all biological pathway categories, and the TRANSFAC database.”

137. Mbatchou, J. et al. Computationally efficient whole-genome regression for quantitative and binary traits. *Nat. Genet.* **53**, 1097–1103 (2021).
183. Zhang, Q.-X. et al. Precise estimation of in-depth relatedness in biobank-scale datasets using deepKin. *Cell Rep. Methods* **5**, 101053 (2025).
144. Raudvere, U. et al. g:Profiler: a web server for functional enrichment analysis and conversions of gene lists (2019 update). *Nucleic Acids Res.* **47**, W191–W198 (2019).
145. Reimand, J., Kull, M., Peterson, H., Hansen, J. & Vilo, J. g:Profiler--a web-based toolset for functional profiling of gene lists from large-scale experiments. *Nucleic Acids Res.* **35**, W193–200 (2007).

Discussion:

“Functional enrichment analysis across CC area, thickness, and volume (from Chen et al¹⁷ and Campbell et al¹⁶ papers) revealed both shared and trait-specific biological signatures (Supplementary Tables 40 and 41). Consistent across all modalities and subregions was strong enrichment of growth factor signaling, particularly PDGFRA/B-driven, PI3K/AKT-related pathways, and RAS-MAP kinase cascade pathways¹⁰⁷. These pathways were significant using the results with and without ICV as a covariate. Growth factor receptor signaling pathways are important intracellular pathways for cellular survival, growth and proliferation through various mechanisms which require further study in the current context^{108–110}. Area measures showed the greatest number of significant associations and the strongest enrichment of these pathways. However, thickness measures, especially in the isthmus, showed the strongest enrichment of these pathways across all morphometry measures and CC subregions. The thickness of isthmus was also the only measure which showed enrichment of the axonemal basal plate, which is crucial

for the formation of cilia and flagella, ensuring proper motility of cells¹¹¹. Volume measures, especially from the Campbell et al.¹⁶ results, highlighted neuronal morphogenesis and guidance pathways - especially in the splenium. These findings suggest that while a core set of signaling pathways influence CC morphology broadly, distinct modalities may reflect different aspects of structural development and cellular regulation.”

2) Related to the first point, since the inclusion of additional metrics is given as the main motivation for an additional CC GWAS, wouldn't it be meaningful to see if there is overlap between the surface area, thickness and volume results, respectively? How does this provide additional mechanistic insight when this is an association study, as are the others? I am still missing an explanation how the differences in the underlying architecture of SA, TH and volume add to our understanding of the relationship with the CC and psychiatric disorders?.

Response: We have conducted a g:Profiler multi-gene-list analysis to determine common/distinct pathways from the area, thickness and volume results from the previous GWAS. We have done this for our results with and without ICV as a covariate. The results are shown in Supplementary Table 40 and 41 (screenshot of Table S40 below). These show the significance of enrichment of gene ontology categories, pathways, and transcription factors across all phenotypes. We have also expanded on the discussion based on these results:

Discussion: *“Functional enrichment analysis across CC area, thickness, and volume (from Chen et al¹⁷ and Campbell et al¹⁶ papers) revealed both shared and trait-specific biological signatures (Supplementary Tables 40 and 41). Consistent across all modalities and subregions was strong enrichment of growth factor signaling, particularly PDGFRA/B-driven, PI3K/AKT-related pathways, and RAS-MAP kinase cascade pathways¹⁰⁷. These pathways were significant using the results with and without ICV as a covariate. Growth factor receptor signaling pathways are important intracellular pathways for cellular survival, growth and proliferation through various mechanisms which require further study in the current context¹⁰⁸⁻¹¹⁰. Area measures showed the greatest number of significant associations and the strongest enrichment of these pathways. However, thickness measures, especially in the isthmus, showed the strongest enrichment of these pathways across all morphometry measures and CC subregions. The thickness of isthmus was also the only measure which showed enrichment of the axonemal basal plate, which is crucial for the formation of cilia and flagella, ensuring proper motility of cells¹¹¹. Volume measures, especially from the Campbell et al.¹⁶ results, highlighted neuronal morphogenesis and guidance pathways - especially in the splenium. These findings suggest that while a core set of signaling pathways influence CC morphology broadly, distinct modalities may reflect different aspects of structural development and cellular regulation.”*

Table S40: Multi-gene-list analysis conducted in gProfiler to test for enrichment of genes mapped to genomic loci identified by corpus callosum area, thickness and volume (previous studies) phenotypes
Enrichment of gene ontology categories, biological pathways and transcription factors identified by genes in the present area and thickness GWAS, and previous volume GWAS of the CC. GO: Gene Ontology, BP: Biological Process, CC: Cellular Component, KEGG: Kyoto Encyclopedia of Genes and Genomes, REAC: F Database. All adjusted p-values within gProfiler were adjusted using the established g-SCS (sets, counts and sizes) threshold.

Source	Term Name	Term ID	Adjusted p value for Area	Adjusted p value for Mean Thickness	Adjusted p value for Volume Chen	Adjusted p value for Volume Campbell
Total						
GO:BP	wound healing, spreading of epidermal cells	GO:0035313	1.00E+00	3.08E-02	1.00E+00	1.00E+00
GO:CC	axon	GO:0030424	1.00E+00	1.00E+00	1.53E-02	3.82E-01
GO:CC	neuron projection	GO:0043005	5.18E-01	1.00E+00	1.73E-02	1.00E+00
GO:CC	phosphatase complex	GO:1903293	1.00E+00	1.00E+00	2.98E-02	1.00E+00
GO:CC	protein serine/threonine phosphatase complex	GO:0008287	1.00E+00	1.00E+00	2.98E-02	1.00E+00
GO:CC	somatodendritic compartment	GO:0036477	1.00E+00	1.00E+00	4.21E-02	1.00E+00
GO:CC	cytoplasmic side of dendritic spine plasma membrane	GO:1990780	1.00E+00	1.00E+00	4.99E-02	4.70E-01
KEGG	mRNA surveillance pathway	KEGG:03015	1.90E-01	1.00E+00	1.87E-02	4.57E-01
REAC	Signaling by cytosolic PDGFRA and PDGFRB fusion proteins	REAC:R-HSA-9673766	9.89E-04	9.83E-05	5.04E-05	3.23E-03
REAC	Diseases of signal transduction by growth factor receptors and second messenger	REAC:R-HSA-5663202	1.52E-02	1.03E-01	3.61E-02	1.89E-03
REAC	Signaling by PDGFR in disease	REAC:R-HSA-9671555	5.00E-02	5.00E-03	2.56E-03	1.82E-01
REAC	Intracellular signaling by second messengers	REAC:R-HSA-9006925	2.72E-03	1.00E+00	6.93E-01	5.92E-01
REAC	Extra-nuclear estrogen signaling	REAC:R-HSA-9009391	1.80E-02	8.50E-02	1.00E+00	1.00E+00
REAC	ESR-mediated signaling	REAC:R-HSA-8039211	1.94E-02	7.32E-01	1.00E+00	1.00E+00
REAC	PI3K activates AKT signaling	REAC:R-HSA-1257604	3.59E-02	9.98E-01	5.19E-01	3.44E-01
REAC	PI3K/AKT Signaling in Cancer	REAC:R-HSA-2219528	3.59E-02	1.35E-01	1.00E+00	1.00E+00
REAC	PI3P, PIP2 and IEG Regulate PI3K/AKT Signaling	REAC:R-HSA-6811508	3.59E-02	1.35E-01	6.95E-02	2.01E-01
REAC	Negative regulation of the PI3K/AKT network	REAC:R-HSA-199418	4.46E-02	1.56E-01	8.05E-02	2.49E-01
TF	Factor: LKLF; motif: NGGGGGG; match class: 1	TF:M05499_1	1.81E-02	1.00E+00	1.00E+00	3.13E-01
TF	Factor: KLF17; motif: NSGGGGG; match class: 1	TF:M05386_1	1.81E-02	1.00E+00	1.00E+00	3.13E-01
Anterior Body						
GO:CC	respiratory chain complex II (succinate dehydrogenase)	GO:0045273	4.99E-03	1.00E+00	1.00E+00	1.00E+00
REAC	Signaling by cytosolic PDGFRA and PDGFRB fusion proteins	REAC:R-HSA-9673766	2.52E-04	5.04E-05	9.83E-05	2.52E-04
REAC	Diseases of signal transduction by growth factor receptors and second messenger	REAC:R-HSA-5663202	1.81E-02	3.61E-02	2.01E-03	1.61E-02
REAC	Signaling by PDGFR in disease	REAC:R-HSA-9671555	1.28E-02	2.56E-03	5.00E-03	1.28E-02
REAC	Disease	REAC:R-HSA-1843885	1.00E+00	1.00E+00	2.18E-02	1.00E+00
REAC	Constitutive Signaling by Aberrant PI3K in Cancer	REAC:R-HSA-2219530	1.00E+00	3.45E-02	1.00E+00	1.00E+00
REAC	Extra-nuclear estrogen signaling	REAC:R-HSA-9009391	2.16E-01	4.37E-02	1.00E+00	1.00E+00
TF	Factor: ZNF775; motif: KGTTTAAGSG; match class: 1	TF:M06597_1	3.76E-03	1.00E+00	1.00E+00	9.45E-03
Genus						
GO:CC	dendrite	GO:0030425	1.00E+00	1.00E+00	1.00E+00	3.15E-02
GO:CC	dendritic tree	GO:0097447	1.00E+00	1.00E+00	1.00E+00	3.20E-02
GO:CC	cytoplasmic side of dendritic spine plasma membrane	GO:1990780	1.00E+00	1.00E+00	4.99E-02	1.45E-01
REAC	Type I heterodisome assembly	REAC:R-HSA-448107	1.00E+00	1.00E+00	5.49E-01	1.49E-02
TF	Factor: ZNF775; motif: KGTTTAAGSG; match class: 1	TF:M06597_1	3.76E-03	1.00E+00	1.00E+00	9.45E-03
Isthmus						
GO:CC	postsynaptic density	GO:0014069	3.69E-02	1.00E+00	1.00E+00	1.00E+00
GO:CC	asymmetric synapse	GO:0032279	4.25E-02	1.00E+00	1.00E+00	1.00E+00
GO:CC	cytosol	GO:0005829	4.48E-02	1.00E+00	1.00E+00	1.00E+00
GO:CC	postsynaptic specialization	GO:0099572	4.71E-02	1.00E+00	1.00E+00	1.00E+00
GO:CC	axonal/lamellar basal plate	GO:0097541	1.00E+00	4.92E-02	1.00E+00	1.00E+00
REAC	Signaling by cytosolic PDGFRA and PDGFRB fusion proteins	REAC:R-HSA-9673766	9.89E-04	9.83E-05	5.04E-05	3.23E-03
REAC	Signaling by PDGFR in disease	REAC:R-HSA-9671555	1.76E-02	1.40E-03	1.40E-03	1.00E+00

3) Thank you for adding more information on the replication results. However, there appears to be an issue with the presented numbers, and they are certainly not in line with the conclusions. It is stated 'We observed a concordance in direction and similar magnitude effect sizes in the analyses within the data from the non-European participants. Out of the 152 significant loci identified across CC phenotypes in this study, 78 loci demonstrated a consistent direction of effect between different populations'. This means the concordance (78/152) is virtually a coin toss. That is not at all reflecting decent generalization across the cohorts. Please comment on this.

Response: We have clarified in the manuscript and in the tables, what we meant by concordance between the European and the non-European ancestries. In the previous tables, concordance was classified as all four cohorts showing the same direction. But the more important effect to show is that the effect of the European cohort is in the 95% confidence interval of the non-European cohort, which was clarified in the same paragraph later in the sentence, and was also the approach used in previous works (Grasby, Jahanshad et al., 2020, *Science*). This better reflects generalization across populations. We have made this statistic the main statistic to show in the manuscript. We have updated the manuscript to highlight this more informative measure as the primary indicator of replication across ancestries. Using this approach, we show 82% of the significant loci in Europeans, generalize to non-Europeans. To help clarify this, we have changed that column in the supplementary tables S1-S2 and S37. We now say how many of the 4 cohorts have the same direction of effect. For example, for total area, genomic locus 4, we show there are 3 cohorts in the positive direction, and 1 cohort in the negative direction.

Manuscript text in Results: "We observed overlaps in the direction and magnitude of effects between the main European analyses with the results from the non-European participants. 124 (82%) out of the 152 significant loci identified across CC phenotypes in this study had effect sizes in European participants falling within the 95% confidence interval of those seen in the non-European participants. Furthermore, 78 loci demonstrated a consistent direction of effect across

all four cohorts (2 European from UKB and ABCD, and 2 non-European from UK Biobank and ABCD, respectively)."

Grasby, K. L., Jahanshad, N., Painter, J. N., Colodro-Conde, L., Bralten, J., Hibar, D. P., Lind, P. A., Pizzagalli, F., Ching, C. R. K., McMahon, M. A. B., Shatokhina, N., Zsembik, L. C. P., Thomopoulos, S. I., Zhu, A. H., Strike, L. T., Agartz, I., Alhusaini, S., Almeida, M. A. A., Alnæs, D., Amlien, I. K., ... Enhancing Neuroimaging Genetics through Meta-Analysis Consortium (ENIGMA)—Genetics working group (2020). The genetic architecture of the human cerebral cortex. *Science (New York, N.Y.)*, 367(6484), eaay6690. <https://doi.org/10.1126/science.aay6690>

4) The response to the feedback on the cohort comparisons/correlations does not fully address how the differences between the cohorts (UKB vs. ABCD) might play a role in the observed low correlations. While age differences are considered, other cohort-specific factors (e.g., ethnicity, environmental influences, or recruitment methods) are likely to contribute to the variability in genetic correlations. You may want to highlight whether these factors have been explored adequately or if they require further attention.

Further, it's true that smaller sample sizes can be a limitation with LDSC. However, while the smaller sample size of the ABCD cohort could influence the results, this is unlikely to account for the observed low correlations. Additionally, if the LDSC method is underpowered, it is worth exploring alternative methods to estimate genetic correlations.

Response: We agree that a myriad of factors can contribute to the lower observed genetic correlations between cohorts. As statistical power can be an issue when computing LDSC on smaller cohorts, we now also conduct all heritability and genetic correlations across cohorts using individual level data across all traits. This was shown to be true as some of the heritability values were very low using LDSC and genetic correlations and high SE, and some values were NA using LDSC for some CC traits. We conducted heritability and genetic correlations using GCTA (reference below) and individual level data. We did this using the *--reml-bivar* option in GCTA while controlling for age, sex and age*sex. Thank you for your suggestion, which resulted in a stronger manuscript.

<https://pubmed.ncbi.nlm.nih.gov/22843982/>

We added the following text to the methods section:

*"To complement these estimates and leverage the availability of individual-level data and greater statistical power, we also employed the bivariate GREML approach in GCTA (--reml-bivar), using the AI-REML algorithm while controlling for age, sex and age*sex, and testing for significance of the genetic correlation against the hypothesis that the genetic correlation is 0 (--reml-bivar-lrt-rg 0)."*

The moderate, but significant genetic correlations across UKB and ABCD for all the CC traits ($r_g = 0.40 - 0.49$) suggest a partially shared genetic architecture of corpus callosum morphology across distinct age groups and populations. As the genetic architecture is not fully overlapping, genetic influences in this case are partially cohort specific. This may signify that genes/genetic

variants may influence the CC differently at different stages of life, or differences may be driven by other cohort-specific differences such as recruitment biases or environmental exposures. Gene expression patterns can also change across the lifespan (i.e. axonal development/migration in childhood vs. myelination in adulthood). GCTA gives higher precision as individual level data are used to estimate heritability and genetic correlations as opposed to summary statistics in LDSC.

Overall these results support a model in which corpus callosum morphology is shaped by a combination of lifespan-persistent and developmentally specific genetic effects. Future work using a GWAS with a larger population of children will be essential to disentangle these effects.

We have added these results to the results section:

“Moderate to high genetic correlations were seen across CC phenotypes between cohorts using LDSC, with r_g ranging from 0.54 (s.e. = 0.27) and 0.92 (s.e. = 0.63) for area metrics, and 0.30 (s.e. = 0.16) and 0.99 (s.e. = 0.69) for thickness metrics. To complement the LDSC approach with an approach using individual level data, we used the bivariate GREML in GCTA⁴⁰. Moderate genetic correlations between cohorts were seen using bivariate GCTA with r_g ranging between 0.40 (s.e. = 0.04) and 0.49 (s.e. = 0.03) across all traits.”

We have added to the discussion to take this into account:

“To better understand the stability of genetic influences on CC morphometry across development, we estimated the genetic correlation of each trait between the UKB (adult) and ABCD (adolescent) cohorts. While initial LDSC-based estimates yielded low/imprecise genetic correlations - likely due to low sample size in ABCD - we complemented this with bivariate GREML using individual-level data via GCTA⁴⁰. GCTA provides more precise estimates by modelling genetic covariance between traits directly and is more robust to smaller sample sizes⁵⁴. We observed consistent and significant genetic correlations across all CC traits ($r_g \approx 0.40-0.49$, Supplementary Table 6), suggesting a partially shared genetic architecture of the CC across the lifespan. While some genetic effects are stable from childhood into adulthood, others may be population specific. The differences could reflect changes in gene expression, shifts in variant effect sizes, or age-dependent neurobiological processes (e.g., neural migration in early life vs. myelination in adulthood)⁵⁵. Environmental and cohort-specific influences are likely to modulate these effects. Future GWASes using a larger sample size of individuals across the lifespan will be crucial to dissect developmental effects and genetic pathways of CC morphometry.”

5) It is still unclear how traits were selected for the genetic correlation analyses. The correlations (and lack thereof for some traits) are still not fully explained.

Response: We selected traits for the genetic correlation analysis based on the availability of sufficiently powered GWAS from disorders and conditions represented as part of ENIGMA working groups. Following the previous submission, we incorporated three additional traits:

Parkinson's disease, insomnia, and substance abuse. All of the traits selected have both active ENIGMA working groups, which study brain structural and functional alterations with the condition, and accessible GWAS summary statistics. We now clarify this selection criteria in the manuscript and have expanded the discussion to explain both the presence and absence of genetic correlations with these traits.

Methods: "Abnormalities of the corpus callosum (CC) have been widely implicated in various neurological and neuropsychiatric conditions. To investigate shared genetic architecture, we selected 22 traits for genetic correlation analyses based on their relevance to CC-related pathology and the availability of well-powered GWAS summary statistics, all of which have active ENIGMA working groups⁷⁸. These traits include Alzheimer's disease (AD)¹⁶⁴, attention deficit-hyperactive disorder (ADHD)⁸⁶, autism spectrum disorder (ASD)¹⁶⁵, anorexia¹⁶⁶, anxiety¹⁶⁷, bipolar disorder¹⁶⁸, chronic overlapping pain conditions¹⁶⁹, depression¹⁷⁰, epilepsy¹⁷¹, intelligence¹⁷², insomnia¹⁷³, neuroticism¹³⁹, obsessive-compulsive disorder (OCD)¹⁷⁴, Parkinson's disease (PD)¹⁷⁵, post-traumatic stress disorder (PTSD)¹⁷⁶, panic disorder¹⁷⁷, schizophrenia¹⁷⁸, substance abuse¹⁷⁹, suicide attempt¹⁸⁰, and Tourette's syndrome¹⁸¹. We used linkage disequilibrium score regression (LDSC) to assess global genetic correlations between these traits and both the area and thickness of the total and parcellated CC. Mendelian randomization and local genetic correlation analyses were then conducted following the same analytic framework used for cortical brain phenotypes."

Based on the addition of the new traits, and the more stringent p-value threshold, the global genetic correlations now have one less significant result for ADHD and area of the anterior body. Furthermore, local genetic correlations show new strong associations with Parkinson's disease and neuroticism, but Tourette's syndrome is no longer significant. This is due to the fact that in the LAVA workflow, common SNPs among all traits from their GWAS summary statistics are used. Based on these new results, we have edited the discussion.

Discussion: "Moreover, all of the local genetic correlations on the 17q21.31 cytogenetic band observed between the genu area and neuroticism, were also observed with PD, but in the positive direction, suggesting heightened risk. The strongest association was with the microtubule-associated protein tau (MAPT) gene. The same locus was also a highly significant sQTL with genu area in the caudate nucleus in the TWAS analysis. This sQTL falls within exon 7 of MAPT, part of the proline rich domain of the tau protein⁹³. This region is known to regulate microtubule binding and aggregation, with the 6p and 6d MAPT isoforms resulting in altered tau assembly and reduced aggregation^{93,94}. Multiple independent studies have linked MAPT to PD risk⁹⁵⁻⁹⁸, and our findings suggest that these splicing events may influence the structural morphology of the genu of the corpus callosum via effects on the caudate nucleus, a region heavily implicated in PD pathology^{99,100}. The genu, which facilitates interhemispheric communication between the prefrontal cortices⁶, shows longitudinal progressive structural decline in PD, with its degeneration correlating with worsening akinetic-rigid motor symptoms¹⁰¹. Lower genu volume has also been reported in PD patients, particularly those with cognitive impairment compared to cognitively intact individuals¹⁰². The loss of dopaminergic neurons in the substantia nigra and the broader nigrostriatal pathway, including within the caudate nucleus,

is a hallmark of PD and is more pronounced in patients with mild cognitive impairment¹⁰³. This finding suggests that altered splicing of MAPT in the caudate nucleus may contribute to increased risk for Parkinson's disease with cognitive impairment through shared genetic effects on the morphometry of the genu, a region critical for facilitating higher-order cognitive functioning.

Several factors can contribute to the lack of global and local genetic correlations of the CC with other tested traits. Our primary explanation is that the CC does not serve a core biological mechanism underlying the pathology of many of the tested traits. Methodological limitations in LDSC and LAVA may also account for a lack of significant results. LDSC requires GWAS summary statistics with a large sample size or a high chi-squared test-statistic. Discordant effects across the genome can also result in null findings. The LAVA workflow uses common SNPs among all the trait's GWAS summary statistics tested, which can reduce the amount of variants used in the analysis. This reduction can decrease the power to detect heritability at each locus, and not meet the stringent multiple comparisons threshold in our analysis ($p < 0.05/18380 = 2.72 \times 10^{-6}$) for a bivariate correlation at each locus to be tested. Moreover, the strict Bonferroni threshold correcting for every test across every trait ($0.05/17909 = 2.79 \times 10^{-6}$) can reduce the likelihood of finding significant bivariate associations. Phenotypic variability and underlying genetic heterogeneity within the tested traits may also contribute to the absence of significant findings. For instance, complex conditions such as Alzheimer's disease^{104,105} and chronic pain¹⁰⁶ are increasingly recognized as comprising multiple subtypes with distinct biological and clinical profiles. Consequently, stratified GWAS analyses, tailored to these subtypes, may be necessary to uncover genetic correlations with our specific neuroanatomical traits."

6) Regarding the feedback of missing info on multiple comparisons correction, the reply is that the Bonferroni corrections are named at each Supplementary Table. This is not sufficient, and it remains still unclear how these corrections were chosen. This is very important. For instance, from the LAVA example included, there appears to be no correction for the number of traits that were tested. This would absolutely be necessary, it cannot be avoided.

Response:

We have revised each relevant Supplementary Table to explicitly state how the Bonferroni correction thresholds were calculated. All SNP data entered into all post-GWAS analysis were based on the corrected threshold of $p < 6.13 \times 10^{-9}$ (i.e., $5 \times 10^{-8} / 8.15$, where 8.15 reflects the number of approximately independent phenotypes after accounting for trait correlations using matrix spectral decomposition). We want to emphasize that for the LAVA analyses, there was always a correction for multiple comparisons. We have now clarified in the text that multiple testing correction was applied across **all pairwise local genetic correlation tests**, spanning **each corpus callosum trait tested against each neuropsychiatric trait**. In the LAVA-TWAS analysis there were a total of 91,791 tests across all CC phenotypes and traits, so the threshold was $0.05/91791 = 5.45 \times 10^{-7}$.

This is for every relevant downstream analysis - including LDSC, LAVA, TWAS and others. We applied stringent multiple testing correction where not only the input data was Bonferroni corrected, but all tests for each analysis were corrected for each test done across all CC traits tested. Depending on the analysis, the input was corrected for all of the traits used in the study.

For example, in the MAGMA analysis, where SNPs were filtered based on the experiment-wide threshold of 6.13×10^{-9} , then further Bonferroni corrections were within each trait's analysis on the FUMA platform. In another case where we ran LDSC partitioned heritability analyses, we show how we calculated the Bonferroni threshold based on every test conducted across all CC traits (experiment-wide). We have made this more clear in every relevant table. We show a few examples from the supplementary table captions below:

Table S3: Results from MAGMA gene-based association analysis for total area of the CC: Results from MAGMA gene-based association analysis. Exome-wide significant (after Bonferroni correction) genes associated with CC phenotypes. Chromosome (CHR), number of SNPs in the genes (N SNPS) and number of relevant parameters used in the model (N PARAM), two-sided association P-values (P) are shown. All GWAS results were input into MAGMA with the experiment-wide Bonferroni threshold correcting for all traits at 6.13×10^{-9} . Bonferroni threshold for each traits analysis was set in MAGMA again at $P = 0.05/18191 = 2.75 \times 10^{-6}$ by correcting for every gene tested.

Table S23: LDSC Partitioned Heritability (Enrichment). RResults of the LDSC-SEG analysis. Prop. SNPs = proportion of SNPs, Prop. h2 = proportion heritability, Prop. h2 std error. Green bolding represents *Bonferroni corrected significance experiment-wide for every trait used in the study* ($0.05/(53*12) = 7.86e-5$).

Table S30: LAVA-TWAS using GTEx-8 Samples (sQTLs): chr = chromosome; start = starting base-pair location of locus; end = ending base-pair location of locus; n.snp = number of SNPs in locus; n.pcs = number of principal components in locus; phen1 = cortical region; phen1_type = surface area (SA) or cortical thickness (CT) of phen1; phen2 = eqtl: expression quantitative trait loci; phen2_type = area or mean thickness of phen2; rho = genetic correlation, rho.lower = lower limit of rho; rho.upper = upper limit of rho; r2 = R-Squared; r2.lower = lower limit of r2; r2.upper = upper limit of r2; p = p-value; gene = gene corresponding to locus; significance threshold was set at the Bonferroni corrected level of $0.05/91791 = 5.45e-07$. *This represents correction for tests run between every sQTL with every CC phenotype. Green bolding represents Bonferroni corrected significance.*

7) The response to the feedback on inclusion of an ICV covariate is minimal. What is the overall observed pattern comparing results with and without ICV, and how is this interpreted?

Response: We redid the entire analysis with ICV as a covariate and showed which genes are common and unique. We also calculated the overlap coefficient of the sets of significant genes in the GWAS with and without ICV. We also calculated the genetic correlation of the results of each trait with and without ICV using LDSC. Supplementary Table 38 shows the differences between the GWAS results, in which significant genes were mapped from significant loci, with and without ICV as a covariate. It also shows the overlap coefficients and LDSC results. We then took these three lists of genes and put them into g:Profiler to do a multi-gene-list analysis

to determine enrichment in genes that were common to both analyses, specific to the no ICV results, and specific to the ICV results. These are shown in Supplementary Table 39.

Methods: *“To determine the degree of overlap of genes in the ICV vs no ICV analysis, the overlap coefficient, or Szymkiewicz–Simpson coefficient was calculated¹⁴¹. In order to determine differences in enrichment in 1) gene ontology categories, 2) biological pathways, and 3) transcription factors, genes mapped to significant loci that were specific to the analysis with or without ICV as a covariate, or common to both, were separately entered into g:Profiler for a multi-gene list analysis¹⁴². All analyses were completed using the g:SCS threshold¹⁴³, all gene ontology categories, all biological pathway categories, and the TRANSFAC database.”*

We have also added an interpretation of these results into the results.

“In order to test for the influence of total intracranial volume (ICV) on the GWAS, another set of GWASes were run controlling for ICV (Methods, Supplementary Table 37). The overlap coefficients and genetic correlations, respectively were 0.64 and 0.75 (s.e = 0.03) for total area and 1 and 1.02 (s.e. = 0.008) for total thickness indicating a high degree of overlap for both analyses. Near perfect genetic correlations were observed comparing across all CC traits, except splenium area ($r_g = 0.08$, s.e. = 0.05), which may be driven by collider bias^{38,39} (Supplementary Table 38). Overall, the strongest enrichment signals were observed among genes that were significant in both the ICV-adjusted and non-ICV GWAS, as well as genes uniquely significant in the non-ICV GWAS - compared to the GWAS that included ICV as a covariate alone. Overall, greater enrichment was observed with genes common between ICV and specific to no ICV, compared to the GWAS which included ICV as a covariate. Genes mapped to significant loci common to both GWAS sets, were consistently mapped to canonical signaling pathways, including PI3K/AKT, PDGFR, and estrogen signaling, suggesting ICV-insensitive mechanisms involved in CC development and maintenance. Genes unique to the no ICV GWAS showed enrichment for mitochondrial respiration, oxidative stress response, and integrin signaling. Genes specific to the ICV-controlled GWAS exhibited much lower enrichment. Regionally specific enrichments were observed for area traits, including WDR5-mediated epigenetic regulation in the genu and apoptosis, as implicated by previous analyses, in the isthmus (Supplementary Table 39).”

As a final general point, please consider to more precisely indicate what is truly revised text in future rebuttals to reviewers. A large proportion of the green text (stated to reflect modified text) was not novel, with perhaps here and there one word in a sentence added. This is misleading. I also find the replies rather brief

Response: We hope our current responses are more clear and precise.